# Innovation of heterochromatin functions drives rapid evolution of essential ZAD-ZNF genes in *Drosophila*

Bhavatharini Kasinathan[1,2,3], Serafin U Colmenares III[4,5,6], Hannah McConnell[3], Janet M Young[3], Gary H Karpen[4,5,6], Harmit S Malik[3,7]*

[1]Medical Scientist Training Program, University of Washington School of Medicine, Seattle, United States; [2]Molecular and Cellular Biology Graduate program, University of Washington School of Medicine, Seattle, United States; [3]Division of Basic Sciences, Fred Hutchinson Cancer Research Center, Seattle, United States; [4]Biological Systems and Engineering Division, Lawrence Berkeley National Laboratory, Berkeley, United States; [5]Department of Molecular and Cell Biology, University of California at Berkeley, Berkeley, United States; [6]Innovative Genomics Institute, Berkeley, United States; [7]Howard Hughes Medical Institute, Fred Hutchinson Cancer Research Center, Seattle, United States

**Abstract** Contrary to dogma, evolutionarily young and dynamic genes can encode essential functions. We find that evolutionarily dynamic *ZAD-ZNF* genes, which encode the most abundant class of insect transcription factors, are more likely to encode essential functions in *Drosophila melanogaster* than ancient, conserved *ZAD-ZNF* genes. We focus on the *Nicknack ZAD-ZNF* gene, which is evolutionarily young, poorly retained in *Drosophila* species, and evolves under strong positive selection. Yet we find that it is necessary for larval development in *D. melanogaster*. We show that *Nicknack* encodes a heterochromatin-localizing protein like its paralog *Oddjob*, also an evolutionarily dynamic yet essential *ZAD-ZNF* gene. We find that the divergent *D. simulans* Nicknack protein can still localize to *D. melanogaster* heterochromatin and rescue viability of female but not male *Nicknack*-null *D. melanogaster*. Our findings suggest that innovation for rapidly changing heterochromatin functions might generally explain the essentiality of many evolutionarily dynamic *ZAD-ZNF* genes in insects.

*For correspondence:
hsmalik@fhcrc.org

Competing interests: The authors declare that no competing interests exist.

## Introduction

Although organisms display enormous phenotypic diversity, their cellular organization and early development are highly conserved across broad taxonomic ranges (*Miklos and Rubin, 1996*). Such widespread conservation has led to a commonly held view that an ancient, conserved genetic architecture encodes fundamental biological functions, which appears to be largely borne out by comparative genomics (*Miklos and Rubin, 1996*; *Krylov et al., 2003*). However, recent studies demonstrated that 30% of 185 evolutionarily young genes in *D. melanogaster* (*Chen et al., 2010*) have acquired roles in development, cell biology, and reproduction that render them essential for viability or fertility (*Lee et al., 2017*; *Long et al., 2013*; *Ross et al., 2013*; *Lee et al., 2019*). Sometimes, evolutionary turnover of genes underlying essential cellular processes can occur, as seen in the evolution of kinetochore proteins (*Drinnenberg et al., 2016*). Even when they are retained over long evolutionary periods, genes encoding essential functions can evolve unexpectedly rapidly across plants and animals (*Malik and Henikoff, 2001*; *Talbert et al., 2004*). Thus, at least a subset

of essential functions is encoded by rapidly evolving genes or genes that are subject to high genetic turnover. The functional basis of this correlation, which runs counter to dogma, is unclear.

To study this unexpected class of rapidly evolving, essential genes further, we focused on the highly dynamic *ZAD-ZNF* gene family, which encodes several essential transcription factors. ZAD-ZNF proteins contain a conserved N-terminal ZAD (Zinc-finger-associated domain), a linker, and a C-terminal domain that includes tandem C2H2 zinc fingers (*Chung et al., 2002*; *Lespinet et al., 2002*). The ZAD facilitates protein-protein interactions but does not have DNA-binding ability, whereas the C2H2 zinc fingers often mediate sequence-specific DNA binding (*Jauch et al., 2003*). Unlike the ZAD and ZNF domains, which are homologous between different ZAD-ZNF proteins, the linker domains are highly variable across ZAD-ZNF proteins in both sequence and length and have no discernible structural motifs. *ZAD-ZNF* genes arose in the ancestor of vertebrates and arthropods, but dramatically expanded within insect lineages (*Chung et al., 2007*), becoming the most abundant class of TFs in many genomes, including in *D. melanogaster* (*Chung et al., 2002*). However, *ZAD-ZNF* gene repertoires can vary quite extensively across insect lineages via gene gains and losses (*Chung et al., 2002*; *Chung et al., 2007*); the causes and consequences of this gene dynamism are poorly understood.

Insect ZAD-ZNF proteins might act analogously to mammalian KRAB-Zinc finger (KZNFs), providing an explanation for their gene dynamism (*Chung et al., 2002*; *Chung et al., 2007*). *KZNF* genes arose in the ancestor of tetrapods and expanded through lineage-specific gene amplifications to become the most abundant class of transcription factors present in mammalian genomes (*Emerson and Thomas, 2009*; *Imbeault et al., 2017*). KZNF proteins contain an N-terminal KRAB (Krüppel-associated box) domain and C-terminal arrays of C2H2 zinc-finger domains that define their DNA-binding specificities. Many *KZNF* genes play a critical role in genome defense through recognition and repression of transposable elements (*Wolf and Goff, 2009*; *Rowe et al., 2010*). As a result, KZNF genes involved in genome defense are subject to rapid genetic turnover and positive selection of their DNA-binding domains (*Thomas and Schneider, 2011*). Similar functions might have driven the diversification of *ZAD-ZNF* genes in insects (*Chung et al., 2002*; *Chung et al., 2007*).

Despite their abundance, only a few *ZAD-ZNFs* have been functionally characterized, with studies restricted to *D. melanogaster*. Approximately half of all *ZAD-ZNF* genes are highly expressed in ovaries and early embryos (*Chung et al., 2002*), where they might play crucial roles for fertility or development. Biochemical characterization of 21 ZAD-ZNF proteins shows that they bind to unique DNA consensus sequences, with putative targets in the regulatory regions of specific target genes (*Krystel and Ayyanathan, 2013*). For example, the ZAD-ZNF protein M1BP (Motif 1 Binding Protein) binds core promoters and promotes the expression of numerous housekeeping genes (*Baumann et al., 2017*; *Li and Gilmour, 2013*). Similarly, *ZAD-ZNF* gene *Grauzone* promotes *cortex* expression and is necessary for meiotic progression during oogenesis (*Chen et al., 2000*; *Harms et al., 2000*; *Page and Orr-Weaver, 1996*). Finally *ZAD-ZNF* genes *Molting Defective, Ouija Board,* and *Séance* encode proteins that promote the transcription of heterochromatin-embedded genes, *Spookier* and *Neverland,* required for larval progression (*Uryu et al., 2018*). In contrast, other ZAD-ZNF proteins repress rather than drive transcription. For example, the ZAD-ZNF protein Oddjob localizes to pericentromeric heterochromatin and is required for gene silencing (position-effect variegation) (*Swenson et al., 2016*). Another ZAD-ZNF protein encoded by *CG17801* helps repress *HetA* and *Blood* transposable elements in the ovary (*Czech et al., 2013*).

Not all ZAD-ZNF functions are directly related to transcription. For example, the ZAD-ZNF proteins ZIPIC, Zw5, and Pita help organize chromatin architecture (*Gaszner et al., 1999*; *Zolotarev et al., 2016*) whereas the ZAD-ZNF protein Trade Embargo mediates the initiation of meiotic recombination during oogenesis (*Lake et al., 2011*). Finally, some ZAD-ZNFs might not function in the nucleus at all. Even though the Weckle ZAD-ZNF protein possesses zinc-finger domains, it localizes to the plasma membrane instead of nuclear chromatin, where it interacts with the Toll-MyD88 complex to help establish the anterior-posterior axis of the developing embryo (*Chen et al., 2006*).

The *ZAD-ZNF* gene family in *Drosophila* is ideal for studying the relationship between genetic innovation and essentiality because of its involvement with essential cellular processes despite rapid evolutionary dynamics. In order to study this relationship between rapid evolution and essentiality, we leveraged extensive phylogenomic and population genetics datasets in *Drosophila* species. We also took advantage of genome-wide screens for phenotype and tools for cytological and genetic

analyses in *D. melanogaster*. Using evolutionary analyses, we identified the *D. melanogaster ZAD-ZNF* genes that had been either subject to genetic turnover or positive selection. Although only a few *ZAD-ZNF* genes have undergone positive selection, we found that these genes are more likely to be required for viability or fertility in *D. melanogaster* than slowly-evolving *ZAD-ZNF* genes. We focused on the characterization of one of these positively-selected *ZAD-ZNF* genes: *Nicknack (CG17802, Nnk)*. We show that *Nicknack* is essential for larval development in *D. melanogaster* despite being evolutionarily young and differentially retained among *Drosophila* species. *Nicknack* belongs to a small cluster of *ZAD-ZNF* paralogs, the best characterized of which is *Oddjob (CG7357, Odj)* (*Swenson et al., 2016*). We show that both *Odj* and *Nnk* encode heterochromatin-localizing proteins. Although Odj broadly localizes to heterochromatin, we found that Nnk predominantly localizes to discrete foci within heterochromatin. Surprisingly, despite a strong signature of positive selection, we found that the protein encoded by the divergent *D. simulans Nicknack* ortholog can still localize to heterochromatin in *D. melanogaster* cells. Furthermore, *D. simulans Nicknack* can significantly rescue the viability of *D. melanogaster Nicknack-null* females but is unable to rescue *Nicknack-null* males. Based on our functional and cytological analyses, we conclude that rapidly changing requirements for heterochromatin function likely drove essential innovation of *ZAD-ZNF* genes such as *Nicknack* and *Oddjob* in *Drosophila*.

## Results

### ZAD-ZNFs genes are dynamic and diverse in *Drosophila*

We searched Flybase to identify all genes in *D. melanogaster* that encode a ZAD domain (PF07776; Pfam database, Pfam.org). We found 91 *ZAD-ZNF* genes distributed across Chromosomes 2, 3, and X. 37 *ZAD-ZNF* genes occur in 13 gene clusters, containing two or more tandemly-arrayed *ZAD-ZNF* genes. Of these, many *ZAD-ZNF* genes share intron/exon structures with their neighbors and likely arose via segmental duplication. In contrast, seven *ZAD-ZNF* genes (*CG3032, CG4318, CG9215, CG44002, CG17361, CG7963, CG17359*) lack introns found in their closest relatives; we infer that these genes were likely born via retrotransposition.

Although ZAD-ZNF proteins are defined as having both a ZAD and a ZNF domain, further analysis of ZAD-containing proteins using NCBI's Conserved Domain Database revealed significant variation in the ZNF domains. We found that ZAD-containing proteins have an average of six C2H2 domains. However, some ZAD-containing proteins have no C2H2 domains at all (*dlip, dbr, CG15435, CG31109, CG31457*), whereas others contain up to 23 C2H2 domains (*CG11902*). In addition to the ZAD in the N-terminus, 13 ZAD-ZNF proteins possess another N-terminal domain, such as a Sodium-Calcium exchanger domain (*CG12391*) or ASF1 histone chaperone domain (*CG10321*). The functional significance of these additional domains is unknown.

Next, we surveyed *ZAD-ZNF* genes in all 12 previously sequenced and annotated *Drosophila* genomes. These species represent a range of evolutionary divergence from *D. melanogaster*, from just a few million years (*e.g., D. simulans*), to more than 40 million years (*e.g., D. virilis*) (*Drosophila 12 Genomes Consortium, 2007*). Our analysis reveals a surprisingly wide range in number of *ZAD-ZNF* genes across different *Drosophila* species genomes (*Figure 1*). For example, we found that the *D. persimilis* genome encodes 130 *ZAD-ZNF* genes whereas the *D. willistoni* genome encodes only 75 *ZAD-ZNF* genes (*Supplementary file 1*). Such analyses are dependent on the state of completion and annotation of individual *Drosophila* species' genomes. Thus, this number may be an underestimate. To complement these analyses, we assessed the apparent age of each *D. melanogaster ZAD-ZNF* gene by examining the presence of orthologs in the 12 annotated genomes of *Drosophila* species (flybase.org). We found that 74 of 91 *D. melanogaster ZAD-ZNF* genes arose in or prior to the common ancestor of all *Drosophila* species, 40 million years ago. Of these 74 *ZAD-ZNF* genes, 61 have been preserved over *Drosophila* evolution, whereas 13 have been lost in at least one lineage or species. We estimate that 17 *ZAD-ZNF* genes arose via gene duplication after the last common ancestor of *D. melanogaster* and *D. virilis*. At least 3 *ZAD-ZNF* genes found in *D. melanogaster* (*CG4318, CG17612, neu2*) originated via gene duplication less than 10 million years ago. Our findings complement previous large-scale surveys that identified rapid changes in *ZAD-ZNF* gene repertoires within insect genomes (*Chung et al., 2002*; *Lespinet et al., 2002*; *Chung et al., 2007*).

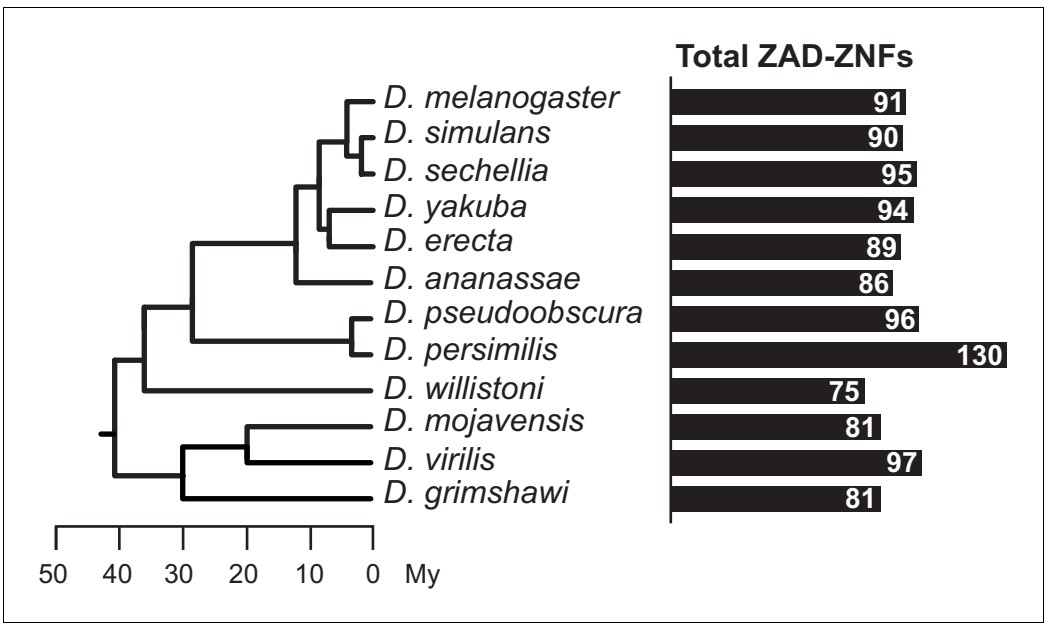

**Figure 1.** Total number of *ZAD-ZNF* genes across *Drosophila*. Phylogeny of 12 *Drosophila* genomes with a scale bar showing approximate divergence times (***Drosophila 12 Genomes Consortium, 2007***). The number of *ZAD*-containing genes in each *Drosophila* species genome (***Supplementary file 1***) is indicated by black bars.

## Rapidly evolving *ZAD-ZNF* genes are frequently essential in *D. melanogaster*

These rapid changes in *ZAD-ZNF* gene repertoires suggested that selection might favor their genetic innovation. We investigated whether evolutionary retention is a predictor of essentiality. We took advantage of the fact that knockdown or knockout phenotypes have been characterized for almost all *D. melanogaster genes.* Indeed, 85 of 91 *D. melanogaster ZAD-ZNF* genes have associated phenotypic outcomes (summarized in ***Supplementary file 1***). Of these, knockdown or knockout of 22 *ZAD-ZNF* genes showed complete lethality or sterility in *D. melanogaster*, whereas the other 63 were determined not to be essential.

Of the 61 *ZAD-ZNF* genes with orthologs retained in all 12 annotated *Drosophila* species' genomes, we found that 14 are essential for either fertility or viability in *D. melanogaster*, whereas 42 genes are not (five genes have no phenotypic data available). In comparison, we found that 8 of the 30 genes not universally conserved in *Drosophila* species are essential, whereas 21 are not (one has no phenotypic data available). Thus, somewhat surprisingly, we find that genes not globally retained over *Drosophila* evolution are just as likely to encode a necessary function in *D. melanogaster* as genes that have been strictly maintained over 40 million years of *Drosophila* evolution (8:21 versus 14:42, p=0.8, two-tailed Fisher's exact test) (***Table 1***). These findings further support the idea that gene families subject to rapid evolutionary turnover may become involved in essential functions.

Gene duplication and loss is only one form of genetic innovation. We also analyzed the *D. melanogaster ZAD-ZNF* genes for signatures of recent positive selection. We performed McDonald-Kreitman tests of *ZAD-ZNF* orthologs found in both *D. melanogaster* and the closely-related species, *D. simulans*. The McDonald-Kreitman test assesses sequence diversity within a species versus divergence between species by comparing the ratio of non-synonymous (amino-acid altering, or replacement) to synonymous substitutions fixed during the divergence of the two species (Dn: Ds), to that of non-synonymous to synonymous polymorphisms within a species (Pn: Ps) (***McDonald and Kreitman, 1991***). The Pn: Ps ratio is a proxy for functional constraint acting on a gene within species and is expected to be similar to Dn: Ds between species under the null hypothesis. However, a higher than expected number of fixed non-synonymous changes would indicate the action of adaptive evolution during species divergence (***McDonald and Kreitman, 1991***).

**Table 1.** Links between *ZAD-ZNF* gene dynamism and essentiality.

(A) *ZAD-ZNF* genes are just as likely to be essential whether or not they are conserved across all 12 *Drosophila* species. Note that five genes for which phenotypic data is not available were not included in these analyses. (B) Positively-selected *ZAD-ZNF*s (via McDonald-Kreitman test, *Table 2*) are more likely to be essential than *ZAD-ZNF*s that have not evolved under positive selection. *D. simulans* lacks one of the 91 *D. melanogaster* genes and no data on essentiality is available for six genes. p-values were calculated by two-tailed Fisher's exact test.

**Conservation of *ZAD-ZNF*s is not correlated with essentiality in *Drosophila***

| *ZAD-ZNF*s | Essential | Not essential | p-value |
|---|---|---|---|
| conserved across *Drosophila* | 14 | 42 | - |
| not conserved across *Drosophila* | 8 | 21 | 0.80 |

**Positively selected *ZAD-ZNF*s are more likely to be essential than *ZAD-ZNF*s not under positive selection in *D. melanogaster***

| *ZAD-ZNF*s | Essential | Not essential | p-value |
|---|---|---|---|
| not under positive selection | 14 | 59 | - |
| positive selection | 8 | 4 | 0.0016 |

We took advantage of previous efforts that sequenced the genomes of hundreds of *D. melanogaster* strains and a reference *D. simulans* strain (*Lack et al., 2016*; *Langley et al., 2012*) to perform the McDonald-Kreitman test using the Popfly server (popfly.uab.cat) (*Hervas et al., 2017*). Of the 91 *D. melanogaster ZAD-ZNF* genes, only *CG2202* is absent in *D. simulans*. We found that 12 out of the remaining 90 *ZAD-ZNF*s show evidence for recent adaptive evolution, *i.e.,* have an excess of fixed non-synonymous changes (Dn) (*Table 2*). Using *D. yakuba* as an outgroup species, we also polarized fixed differences between *D. melanogaster* and *D. simulans* to assess whether the lineage leading to *D. melanogaster* showed evidence of positive selection in these 12 *ZAD-ZNF* genes. We found that 5 out of 12 genes showed evidence of positive selection with this polarized McDonald-Kreitman test (*Table 2*).

Subsequently, we used a domain-restricted analysis to define which of the three protein domains (ZAD, linker, or ZNF) were subject to positive selection. Although we found evidence of domain-specific positive selection in eight cases (two in the case of the polarized McDonald-Kreitman test), we were unable to define the domain subject to positive selection in four cases, due to small numbers of intra-species polymorphisms (*Table 2*). One gene (*CG7386*) showed signatures of adaptive evolution in its ZAD domain, whereas three genes (*CG2712, CG7386, CG10321*) showed evidence of adaptive evolution in the ZNF domain. Unexpectedly, we found that the linker domain has evolved under positive selection in six *ZAD-ZNF* genes. The biochemical function of these linker domains is largely unexplored as they lack predicted structural motifs (*Chung et al., 2002*; *Chung et al., 2007*), are highly variable in length and sequence even between orthologs, and are predicted to encode intrinsically disordered domains (*Ishida and Kinoshita, 2007*). Our findings implicate the poorly characterized linker region in mediating the adaptive potential of several *ZAD-ZNF* genes.

Using this signature of positive selection in a subset of *ZAD-ZNF* genes, we evaluated whether positively selected genes encode essential functions. Intriguingly, we found that 8 of 12 *ZAD-ZNF* genes that have evolved under positive selection are essential, whereas only 14 of the remaining 73 genes that have been phenotypically assayed are essential (six have no phenotypic data available). Thus, contrary to the dogma, we find that *ZAD-ZNF* genes that are subject to positive selection are more likely to be essential for viability or fertility (8:4 versus 14:59, two-tailed Fisher's exact test, p=0.0016) (*Table 1*). Our findings not only imply that several essential *ZAD-ZNF* genes evolve rapidly, but also raise the possibility that rapid evolution of some *ZAD-ZNF* genes might be critical for organismal viability.

**Table 2.** Positively selected *ZAD-ZNF*s.

Summary statistics for the McDonald-Kreitman (MK) test (with statistically significant values at p<0.05). The number of polymorphisms within *D. melanogaster* at synonymous (Ps) and non-synonymous (Pn) sites are compared to the number of all fixed synonymous (Ds) and non-synonymous (Dn) changes between *D. melanogaster* and *D. simulans*. A polarized MK test only considers synonymous (Ds) and non-synonymous (Dn) changes that were fixed along the lineage leading to *D. melanogaster*. MK test statistics were calculated either for the whole gene or individual domains. A neutrality index (N.I.) of <1 suggests an excess of fixed non-synonymous changes between species (i.e., positive selection). Phenotypes of these genes are taken from sources that are listed in ***Supplementary file 1***.

| ZAD-ZNF gene (mutant phenotype) | Region | Alignment length (codons) | McDonald-Kreitman test (unpolarized) | | | | | | Polarized changes (*D. melanogaster*) | | | | Polarized changes (*D. simulans*) | | Total polarized Dn | Total polarized Ds |
|---|---|---|---|---|---|---|---|---|---|---|---|---|---|---|---|---|
| | | | p-value | Dn | Ds | Pn | Ps | NI | p-value | Dn | Ds | NI | Dn | Ds | | |
| Nnk (CG17802) (lethal) | full-length | 448 | **0.007** | 52 | 24 | 14 | 20 | 0.323 | **0.041** | 17 | 8 | 0.329 | 26 | 16 | 43 | 24 |
| | ZAD | 71 | 0.119 | 6 | 2 | 2 | 4 | 0.167 | 0.058 | 3 | 0 | 0.000 | 3 | 2 | 6 | 2 |
| | linker | 216 | **0.019** | 34 | 10 | 9 | 10 | 0.265 | 0.070 | 11 | 3 | 0.245 | 15 | 7 | 26 | 10 |
| | C2H2 | 136 | 0.091 | 6 | 11 | 0 | 6 | 0.000 | 0.155 | 2 | 5 | 0.000 | 4 | 6 | 6 | 11 |
| Odj (CG7357) (lethal) | full-length | 438 | **0.000** | 40 | 32 | 5 | 25 | 0.160 | **0.002** | 18 | 15 | 0.167 | 16 | 17 | 34 | 32 |
| | ZAD | 73 | 0.197 | 4 | 4 | 0 | 2 | 0.000 | 0.248 | 1 | 1 | 0.000 | 3 | 3 | 4 | 4 |
| | linker | 138 | 0.053 | 18 | 9 | 4 | 8 | 0.250 | 0.309 | 5 | 4 | 0.400 | 9 | 5 | 14 | 9 |
| | C2H2 | 135 | 0.099 | 2 | 7 | 0 | 11 | 0.000 | | 0 | 4 | | 2 | 3 | 2 | 7 |
| Trem (CG4413) (female sterile) | full-length | 442 | **0.015** | 31 | 30 | 3 | 14 | 0.207 | 0.335 | 6 | 13 | 0.464 | 21 | 17 | 27 | 30 |
| | ZAD | 77 | 0.571 | 2 | 6 | 0 | 1 | 0.000 | | 0 | 1 | | 2 | 5 | 2 | 6 |
| | linker | 192 | **0.036** | 18 | 13 | 2 | 8 | 0.181 | 0.407 | 4 | 7 | 0.438 | 13 | 6 | 17 | 13 |
| | C2H2 | 133 | 0.239 | 3 | 10 | 0 | 5 | 0.000 | 0.338 | 1 | 5 | 0.000 | 2 | 5 | 3 | 10 |
| Zw5 (CG2711) (lethal) | full-length | 590 | **0.004** | 55 | 51 | 6 | 22 | 0.253 | 0.120 | 15 | 23 | 0.418 | 28 | 28 | 43 | 51 |
| | ZAD | 78 | 0.125 | 3 | 4 | 0 | 4 | 0.000 | 0.285 | 1 | 3 | 0.000 | 1 | 1 | 2 | 4 |
| | linker | 229 | 0.251 | 32 | 17 | 3 | 4 | 0.398 | 0.867 | 7 | 8 | 0.857 | 16 | 9 | 23 | 17 |
| | C2H2 | 217 | 0.060 | 12 | 25 | 0 | 8 | 0.000 | 0.159 | 3 | 11 | 0.000 | 7 | 14 | 10 | 25 |
| CG2712 (viable) | full-length | 524 | **0.003** | 70 | 36 | 15 | 24 | 0.321 | 0.210 | 21 | 19 | 0.565 | 37 | 15 | 58 | 34 |
| | ZAD | 80 | 0.161 | 10 | 6 | 3 | 6 | 0.300 | 0.569 | 2 | 2 | 0.500 | 7 | 4 | 9 | 6 |
| | linker | 231 | 0.089 | 45 | 14 | 10 | 8 | 0.389 | 0.106 | 16 | 4 | 0.313 | 18 | 8 | 34 | 12 |
| | C2H2 | 190 | **0.038** | 15 | 14 | 2 | 10 | 0.187 | 0.759 | 3 | 11 | 0.733 | 12 | 3 | 15 | 14 |
| D19B (CG10270) (viable) | full-length | 774 | **0.005** | 11 | 42 | 3 | 66 | 0.174 | **0.049** | 4 | 20 | 0.227 | 5 | 21 | 9 | 41 |
| | ZAD | 72 | 0.121 | 2 | 4 | 0 | 6 | 0.000 | 0.134 | 1 | 2 | 0.000 | 1 | 2 | 2 | 4 |
| | linker | 171 | 0.182 | 6 | 12 | 2 | 13 | 0.308 | 0.347 | 0 | 6 | | 4 | 5 | 4 | 11 |
| | C2H2 | 257 | | 0 | 6 | 0 | 25 | | | 0 | 3 | | 0 | 3 | 0 | 6 |
| | C2H2-2 | 78 | | 0 | 2 | 0 | 7 | | | 0 | 2 | | 0 | 0 | 0 | 2 |

*Table 2 continued on next page*

*Table 2 continued*

| ZAD-ZNF gene (mutant phenotype) | Region | Alignment length (codons) | McDonald-Kreitman test (unpolarized) | | | | | | Polarized changes (*D. melanogaster*) | | | | Polarized changes (*D. simulans*) | | Total polarized Dn | Total polarized Ds |
|---|---|---|---|---|---|---|---|---|---|---|---|---|---|---|---|---|
| | | | p-value | Dn | Ds | Pn | Ps | NI | p-value | Dn | Ds | NI | Dn | Ds | | |
| CG7386 (lethal) | full-length | 688 | 0.002 | 78 | 38 | 28 | 36 | 0.379 | 0.117 | 26 | 18 | 0.538 | 47 | 19 | 73 | 37 |
| | ZAD | 72 | 0.858 | 3 | 6 | 3 | 5 | 1.200 | 0.506 | 1 | 4 | 2.400 | 1 | 2 | 2 | 6 |
| | linker | 116 | 0.118 | 12 | 5 | 13 | 1 | 5.417 | 0.133 | 4 | 2 | 6.500 | 7 | 3 | 11 | 5 |
| | C2H2-1 | 21 | | 0 | 0 | 0 | 1 | | | 0 | 0 | | 0 | 0 | 0 | 0 |
| | C2H2-2 | 163 | 0.005 | 14 | 7 | 5 | 16 | 0.156 | 0.639 | 2 | 4 | 0.625 | 12 | 3 | 14 | 7 |
| | C2H2-3 | 78 | 0.098 | 2 | 6 | 3 | 1 | 9.000 | 0.540 | 1 | 1 | 3.000 | 0 | 5 | 1 | 6 |
| CG17359 (lethal) | full-length | 344 | 0.000 | 43 | 17 | 3 | 14 | 0.085 | 0.000 | 18 | 6 | 0.071 | 20 | 11 | 38 | 17 |
| | ZAD | 82 | 0.157 | 3 | 1 | 1 | 3 | 0.111 | 0.540 | 1 | 1 | 0.333 | 1 | 0 | 2 | 1 |
| | linker | 134 | 0.000 | 28 | 4 | 1 | 5 | 0.029 | 0.001 | 9 | 0 | 0.000 | 15 | 4 | 24 | 4 |
| | C2H2 | 105 | 0.064 | 10 | 10 | 0 | 4 | 0.000 | 0.040 | 6 | 4 | 0.000 | 4 | 6 | 10 | 10 |
| *wek (CG4148)* (lethal; maternal effect lethal) | full-length | 473 | 0.001 | 29 | 42 | 3 | 30 | 0.145 | 0.596 | 3 | 19 | 0.633 | 16 | 23 | 19 | 42 |
| | ZAD | 70 | 0.206 | 2 | 3 | 0 | 3 | 0.000 | | 0 | 3 | | 1 | 0 | 1 | 3 |
| | linker | 194 | 0.012 | 22 | 19 | 3 | 14 | 0.185 | 0.456 | 3 | 7 | 0.500 | 11 | 12 | 14 | 19 |
| | C2H2 | 157 | 0.074 | 5 | 17 | 0 | 12 | 0.000 | | 0 | 6 | | 4 | 11 | 4 | 17 |
| CG10321 (viable) | full-length | 835 | 0.001 | 28 | 50 | 9 | 63 | 0.255 | 0.007 | 11 | 20 | 0.260 | 14 | 29 | 25 | 49 |
| | ZAD | 69 | | 0 | 2 | 0 | 4 | | | 0 | 0 | | 0 | 2 | 0 | 2 |
| | linker | 546 | 0.008 | 24 | 38 | 8 | 42 | 0.302 | 0.072 | 8 | 15 | 0.357 | 13 | 22 | 21 | 37 |
| | C2H2 | 146 | 0.024 | 3 | 7 | 0 | 15 | 0.000 | 0.010 | 2 | 3 | 0.000 | 1 | 4 | 3 | 7 |
| *mld (CG34100)* (lethal) | full-length | 2021 | 0.020 | 84 | 70 | 37 | 57 | 0.541 | 0.328 | 32 | 36 | 0.730 | 37 | 32 | 69 | 68 |
| | ZAD | 70 | | 0 | 0 | 0 | 0 | | | 0 | 0 | | 0 | 0 | 0 | 0 |
| | linker | 1143 | 0.282 | 63 | 51 | 30 | 34 | 0.714 | 0.824 | 24 | 25 | 0.919 | 29 | 24 | 53 | 49 |
| | linker-2 | 278 | 0.075 | 6 | 8 | 2 | 13 | 0.205 | 0.292 | 2 | 4 | 0.308 | 2 | 4 | 4 | 8 |
| | C2H2-1 | 81 | 0.237 | 4 | 5 | 0 | 2 | 0.000 | 0.290 | 2 | 3 | 0.000 | 2 | 2 | 4 | 5 |
| | C2H2-2 | 110 | 0.427 | 1 | 1 | 1 | 4 | 0.250 | 0.121 | 1 | 0 | 0.000 | 0 | 1 | 1 | 1 |
| CG4282 (viable) | full-length | 652 | 0.030 | 13 | 32 | 2 | 25 | 0.197 | 0.061 | 6 | 16 | 0.213 | 8 | 15 | 14 | 31 |
| | ZAD | 75 | 0.053 | 2 | 0 | 1 | 4 | 0.000 | 0.121 | 1 | 0 | 0.000 | 1 | 0 | 2 | 0 |
| | linker | 248 | 0.252 | 9 | 18 | 1 | 7 | 0.286 | 0.433 | 4 | 11 | 0.393 | 6 | 6 | 10 | 17 |
| | C2H2 | 290 | | 0 | 11 | 0 | 14 | | | 0 | 3 | | 0 | 8 | 0 | 11 |

## The *Oddjob-Nicknack* cluster of *ZAD-ZNF* genes evolves dynamically in *Drosophila*

To further investigate the biological basis for the correlation between positive selection and gene essentiality, we decided to focus on one cluster of *ZAD-ZNF* genes on chromosome 3 of *D. melanogaster* (*Figure 2A*). This cluster of five genes contains two of the eight positively-selected, essential genes in *D. melanogaster*: *Oddjob (Odj, CG7357)* and *CG17802*; both also show evidence of positive selection in the polarized McDonald-Kreitman test (*Table 2*). *Odj is the* best-characterized gene in this cluster. The Oddjob protein interacts with Heterochromatin Protein 1a (HP1a),

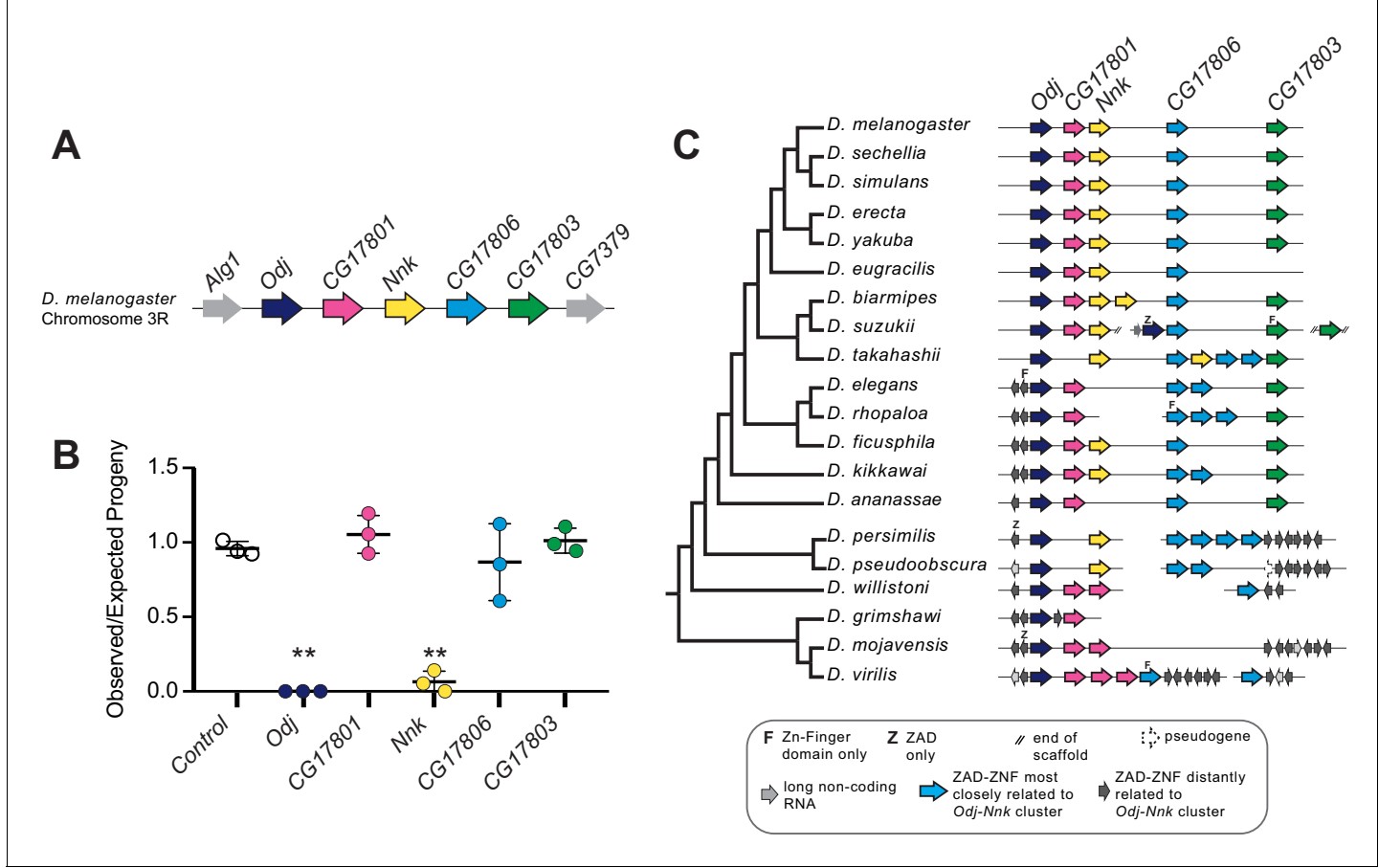

**Figure 2.** *Nicknack* and *Oddjob* are essential for viability. (**A**) A schematic of the *Oddjob-Nicknack* cluster of *ZAD-ZNF* genes in *D. melanogaster.* (**B**) Viability of adult flies ubiquitously knocked down for each *Odj-Nnk* cluster member. The vertical axis shows a ratio of the observed/expected number of knockdown progeny per cross; each cross is represented by a point. Ubiquitous knockdown of either *Odj* or *Nnk* greatly reduces adult viability. Horizontal bars represent the mean and one standard deviation. We compared controls and *ZAD-ZNF* knockdowns using a two-tailed Student's t-test; ** denotes p-value<0.01. (**C**) Phylogenomic analysis of the *Odj-Nnk* cluster shows that *Oddjob*, *CG17801* and *CG17806* all date back to the origin of the *Drosophila* genus. However, *Oddjob* is the only gene within the cluster retained in all queried *Drosophila* species; *CG17801* has been lost within the *obscura* clade and in *D. takahashii*, whereas *CG17806* has undergone numerous gains and losses. *CG17803* arose in the ancestor of *D. melanogaster* and *D. ananassae* but was lost in *D. eugracilis*. *Nicknack* arose in the ancestor of *D. melanogaster* and *D. pseudoobscura*, duplicated in *D. biarmipes* and *D. takahashii,* but was independently lost in *D. ananassae* and the *D. elegans/D. rhopaloa* lineage.

The online version of this article includes the following figure supplement(s) for figure 2:

**Figure supplement 1.** Phylogenetic analysis of all genes found within the *Odj-Nnk ZAD-ZNF* cluster across 20 *Drosophila* species based on a multiple alignment of their ZNF domains (*Supplementary file 2*).

---

dynamically localizes to heterochromatin, and is required for heterochromatin-mediated gene silencing, or position-effect variation (*Swenson et al., 2016*). Knockdown or over-expression of *Odj* is lethal in *D. melanogaster*, although the functional basis for this lethality is still unknown (*Schertel et al., 2015*). In keeping with the *Oddjob* nomenclature theme of 'James Bond henchmen,' and the mutant phenotype (described below), we renamed *CG17802* as *Nicknack*, or *Nnk*. This cluster of *ZAD-ZNF* genes also includes three genes that do not evolve under positive selection (*CG17801, CG17806*, and *CG17803*) in *D. melanogaster* (*Figure 2A*). Although previous studies found that the knockdown of *CG17801* in the germline led to de-repression of HetA and Blood transposable elements in the ovary, *CG17801* knockdown did not significantly impair fertility or viability (*Czech et al., 2013*). The other two *ZAD-ZNF* genes in this cluster (*CG17803, CG17806*) have not been previously investigated.

We evaluated the functional roles of all five *ZAD-ZNF* genes in the *Odj-Nnk* cluster by generating knockdowns using an Actin5C-Gal4 driver and RNAi constructs specific to each gene (*Figure 2B*) to generate ubiquitous knockdown of individual genes in the resulting zygotes. Consistent with previous results (*Schertel et al., 2015*), we found that the ubiquitous knockdown of *Odj* or *Nnk* is lethal. In contrast, we found that knockdowns of *CG17801*, *CG17803* and *CG17806* are viable (*Figure 2B*). Based on these results, we conclude that the two positively-selected members of this *ZAD-ZNF* cluster – *Odj* and *Nnk* – are both essential, confirming the unexpected correlation we previously observed between positive selection and gene essentiality in the *ZAD-ZNF* gene family (*Table 1B*).

To gain deeper insight into the evolutionary dynamics of the *Odj-Nnk* cluster, we identified orthologs of these genes using reciprocal TBLASTN searches with each of the five *D. melanogaster* genes as queries. We searched both the originally-sequenced, well-annotated 12 *Drosophila* genomes (*Drosophila 12 Genomes Consortium, 2007*), as well as eight additional genomes that were subsequently sequenced to sample the melanogaster group within the *Sophophora* subgenus more densely (*Chen et al., 2014*). In several cases, we were not able to confidently assign the TBLASTN hits to orthologous groups because they matched closely to more than one *D. melanogaster* gene, or there were several putative hits within a single genome, or because the hit contained only the ZAD or ZNF domain.

To more rigorously identify orthologs, we conducted phylogenetic analyses (*Figure 2—figure supplement 1*) using a multiple alignment of the ZNF domain (*Supplementary file 2*). Our phylogenomic analyses reveal that *Odj-Nnk* cluster evolution was highly dynamic during the evolution of the *Drosophila* genus (summarized in *Figure 2C*). Although *Oddjob* orthologs are present throughout 40 million years of *Drosophila* evolution, no other member of the *D. melanogaster Odj-Nnk* cluster is universally present in *Drosophila* species. In addition to *Odj*, *CG17801* and *CG17806* also date back prior to the origin of *Drosophila* but unlike *Odj*, *CG17801* and *CG17806* have since been lost in some species (*Figure 2C*). While *CG17801* has been lost in the *obscura* group,*CG17806* underwent multiple independent duplication and loss events. *CG17803* arose in the ancestor of *D. melanogaster* and *D. ananassae* and underwent two independent losses. Finally, *Nnk* appears to have arisen in the ancestor of *D. melanogaster* and *D. pseudoobscura* (~30 mya, *Figure 1*), and later experienced multiple independent duplications and losses (*Figure 2C*). We note that our estimates for the age of *Nnk* are higher than those reported previously (*Chen et al., 2010*), likely because the prior estimate was based on fewer sequenced species. Based on these analyses, we conclude that, despite its essential function in *D. melanogaster*, *Nnk* is not universally conserved in *Drosophila* species.

## *Nicknack* is an essential *ZAD-ZNF* gene in *D. melanogaster*

*Nnk* has a dramatic evolutionary history: young evolutionary age, differential retention, positive selection. Yet, it serves an essential function in *D. melanogaster* based on our and other previous analyses. However, this claim of essentiality has been challenged by two previous findings. First, the screen that originally identified *Nnk* as an essential gene used the KK RNA-interference (RNAi) collection from the Vienna *Drosophila* Stock Center (VDRC) (*Figure 3A*; *Chen et al., 2010*). Many lines in this 'KK' collection were later found to harbor a second-site mutation that caused lethality as a result of ectopic *tiptop* expression when crossed to *GAL4*-driver lines (*Green et al., 2014*; *Vissers et al., 2016*). As a result of its dependence on the KK lines, the claim of *Nnk* essentiality remained ambiguous. A second *Nnk* mutant allele, created by CRISPR-Cas9-mediated mutagenesis (hereafter referred to as *CRISPR-null*), had a four base pair deletion within the coding sequence of the gene that created a frameshift and a premature stop codon within the linker domain of *Nnk* (*Kondo et al., 2017*; *Figure 3A*). Although this deletion was homozygous lethal, it was unexpectedly viable when paired with a deficiency covering this *ZAD-ZNF* cluster (*Kondo et al., 2017*), again challenging the result that *Nnk* is an essential gene.

Given these ambiguous results and the importance of *Nnk* for our claims of *ZAD-ZNF* essentiality despite evolutionary innovation, we re-investigated whether *Nnk* is essential for viability in *D. melanogaster*. We found several lines of evidence to support the conclusion that it is indeed essential (*Figure 3B–D*). First, we began by validating the RNAi line used in the original study (*Chen et al., 2010*), which first identified *Nnk* as a young, essential gene. We found that the *Nnk* VDRC RNAi line does not have an insertion upstream of the *tiptop* gene, which is associated with lethality in other KK lines (*Green et al., 2014*; *Vissers et al., 2016*). Second, we were able to rescue this lethality via

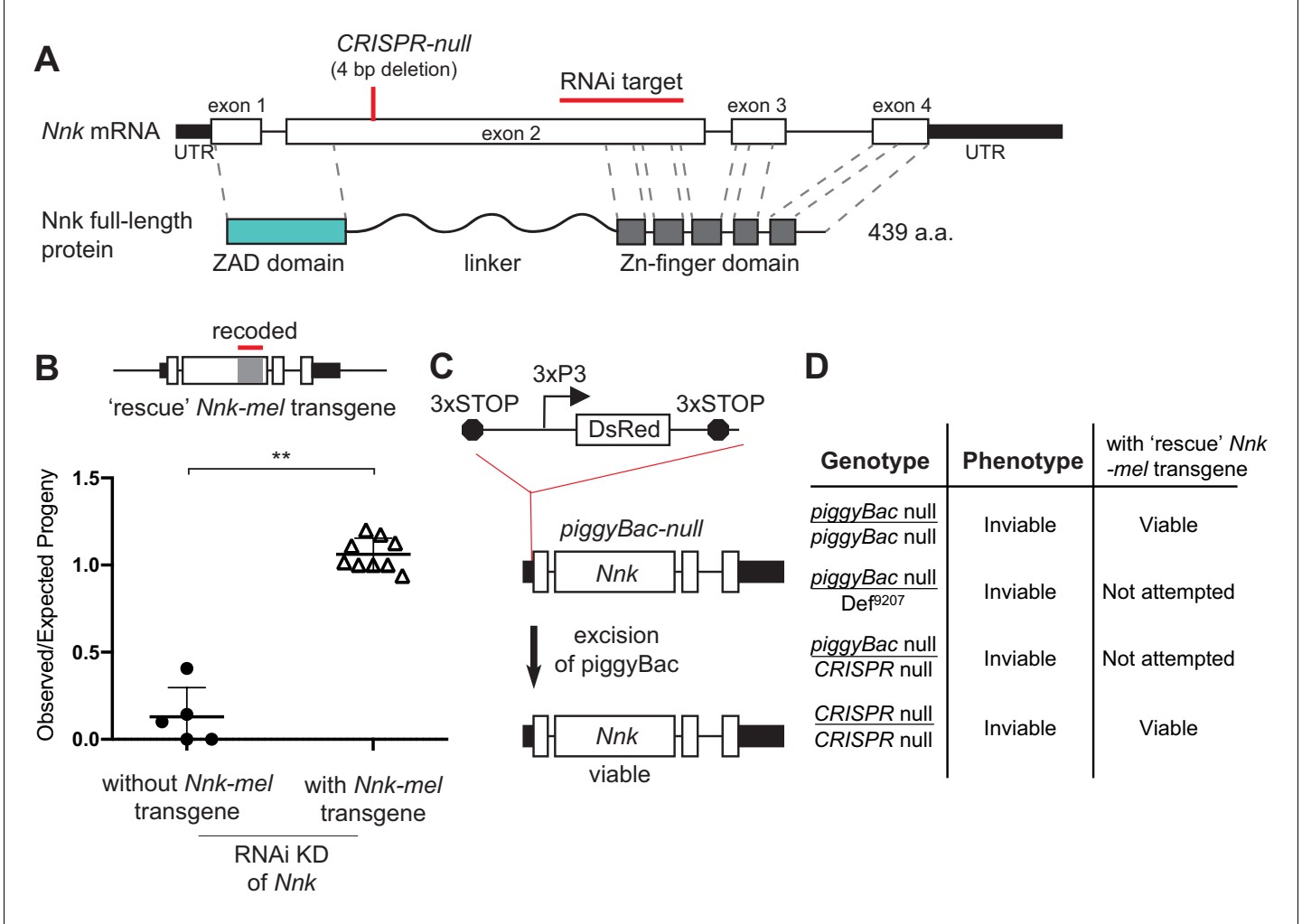

**Figure 3.** *Nicknack* RNAi and mutant alleles. (A) Schematic of *Nnk* gene. The *Nnk* gene contains four coding exons and encodes a full-length protein containing ZAD, linker, and ZNF domains. The CRISPR-null is a 4 bp deletion within exon 2 (*Kondo et al., 2017*): this deletion creates a frameshift and a premature stop codon within the linker region. The VDRC RNAi KK line creates double-stranded RNA corresponding to a region of exon 2 (schematized). (B) Schematic of *Nnk-rescue* transgene design, which contains the genomic region 1 kb upstream of the *Nnk* start codon and 1 kb downstream of the stop codon. The region targeted by the RNAi construct (highlighted) was recoded by synonymous mutations in the *Nnk*-rescue transgene to make it resistant to RNAi. Upon ubiquitous knockdown of *Nnk* using Act5c-GAL4 driven RNAi, adult viability is very low, but is significantly restored by the RNAi-resistant *Nnk*-rescue transgene. Horizontal bars represent the mean and error bars represent one standard deviation of replicates of single-pair matings for each cross. Vertical axis represents the observed/expected number of progeny from each cross. Asterisks indicate p<0.01, two-tailed Student's t-test. (C) *piggyBac*-null is a *piggyBac* insertion in the 5' UTR of *Nnk* that contains a fluorescent reporter (DsRed) driven by an eye specific promoter (3xP3) and flanked by stop codons in all three reading frames which terminate translation through *Nnk* (*Schuldiner et al., 2008*). Excision of this *piggyBac* insertion restores viability. (D) Allelic combinations of CRISPR/*piggyBac*-nulls/genetic deficiencies were tested, in some cases in the presence of the *Nnk*-rescue transgene, to investigate and confirm *Nnk* essentiality (*Figure 3—source data 1*).

The online version of this article includes the following source data for figure 3:

**Source data 1.** Various *Nnk* disruption alleles can be all rescued significantly with a *Nnk-mel* rescue transgene.

complementation with an intact *Nnk-mel* rescue transgene. This *Nnk-mel* rescue transgene was flanked by regulatory sequences from the endogenous locus (1 kb segments upstream and downstream of endogenous *Nnk*) and recoded via mutations in synonymous sites to make it resistant to RNAi knockdown (*Figure 3B*; *Supplementary file 4*). Endogenous levels of *Nnk* expression are too low for us to validate expression of the transgene. However, our ability to rescue *Nnk* knockdown-mediated inviability strongly imply that the recoded transgene is appropriately expressed (*Figure 3B*).

To rule out any indirect effects arising from RNAi knockdown, we also examined two previously generated mutant alleles of *Nnk*. The first is a *piggyBac* insertion two bp upstream of the start codon in the 5′ UTR of *Nnk* (*Schuldiner et al., 2008*; *Figure 3C*). This *piggyBac* insertion, which is marked by a DsRed reporter driven by an eye-specific promoter, is flanked by stop codons in all three reading frames, which prevents translation downstream of the insertion. We refer to this insertion as a *piggyBac*-null allele of *Nnk*. We found that this insertion allele is homozygous lethal but viability can be fully rescued upon mobilization of the *piggyBac* element, which repairs the intact 5′ UTR in a 'scarless' fashion to restore *Nnk* function (schematized in *Figure 3C*). Furthermore, *piggyBac*-null flies can also be rescued by the *Nnk*-rescue transgene (*Figure 3D*; *Figure 3—source data 1*). Second, we re-examined the previously-generated *CRISPR*-null *Nnk* allele (*Kondo et al., 2017*; *Figure 3A*). Contrary to previous results, we found that this allele is lethal when paired with a *Nnk*-spanning deficiency (BL9207, *Figure 3D*) and also fails to complement the *piggyBac*-null allele (*Figure 3D*; *Figure 3—source data 1*; *Supplementary file 4*). Moreover, the *Nnk*-rescue transgene can restore the viability of the *CRISPR*-null *Nnk* allele (*Figure 3D*; *Figure 3—source data 1*). Based on all these results, we conclude that *Nnk* is unambiguously an essential gene in *D. melanogaster*.

## *Nicknack* is required for larval development in *D. melanogaster*

Next, we investigated the developmental stage at which *Nnk-null* progeny die. For this, we crossed *Nnk-null* heterozygotes to each other (*Figure 4A*). These flies contain the *piggyBac* insertion upstream of the *Nnk* gene (*Figure 3C*) on one chromosome, along with a balancer chromosome (TM3G) marked with *GFP* and carrying a wildtype *Nnk* allele. Progeny homozygous for the balancer chromosome TM3G die as early embryos. Consequently, all larvae lacking *GFP* expression are *Nnk-null* homozygotes whereas those that are heterozygous express the *GFP* encoded on the balancer. We conducted egg-lay experiments from crosses between heterozygote *Nnk-null* flies and tracked the developmental progression of *GFP*-negative, *Nnk-null* progeny relative to their *GFP*-expressing heterozygote siblings. We found that *Nnk-null* progeny progress through embryogenesis at the same rate as their heterozygote siblings and are morphologically indistinguishable from heterozygotes until the L1 larval stage. The *Nnk-null* larvae are able to move toward and consume yeast paste much like their heterozygote siblings. However, when heterozygote siblings molt into the L2 stage 48 hr after egg laying (AEL), *Nnk-null* larvae do not molt (*Figure 4B*). Instead, 48 hr AEL, *Nnk-null* larvae progressively become unable to move or eat, eventually dying by 60 hr AEL.

The *Nnk-null* larval arrest phenotype is reminiscent of previous findings with the *ZAD-ZNF* genes *Séance*, *Ouija Board* and *Molting Defective*, which encode proteins necessary for expression of the *Spookier* and *Neverland* genes required for ecdysone biosynthesis (*Uryu et al., 2018*; *Neubueser et al., 2005*; *Komura-Kawa et al., 2015*). Defects in any of the *Séance*, *Ouija Board* and *Molting Defective ZAD-ZNF* genes leads to arrest of larval development and death (*Uryu et al., 2018*; *Neubueser et al., 2005*; *Komura-Kawa et al., 2015*). However, this lethality can be rescued by supplementing the diet with ecdysone or by overexpression of ecdysone biosynthetic enzymes, bypassing the requirement for these three *ZAD-ZNF* genes. We therefore, tested whether dietary supplementation with an ecdysone precursor, cholesterol (7DH, or 7-Dehydrocholesterol) or ecdysone (20E, or 20-Hydroxyecdysone) could bypass or significantly delay the death of *Nnk-null* larvae (*Figure 4C*). We found that it could not. In contrast, the same dietary supplementation is able to significantly restore the viability of *Npc1a* larvae, which lack an essential transporter of cholesterol (*Figure 4C*). Based on these results, we conclude that *Nnk* plays an essential role in larval progression in *D. melanogaster* that is distinct from known steps of the ecdysone biosynthesis pathway.

## Transcriptional consequences of *Nicknack* knockout in *D. melanogaster* larvae

To further investigate our developmental findings, we also performed RNA-seq analyses, comparing the transcriptomes of *Nnk-null* larvae to their age-matched heterozygote siblings. We compared the two genotypes at each of two important time-points. First, we compared transcriptomes 24 hr AEL, when the two genotypes are morphologically indistinguishable L1 larvae. At this timepoint, we find that 249 genes (2.4% of all expressed genes) are differentially expressed between *Nnk-null* larvae and control larvae, with 116 genes at least two-fold upregulated and 133 genes two-fold downregulated in *Nnk-null* larvae (*Figure 4D*). Among genes downregulated in *Nnk-null* L1 larvae, we find a

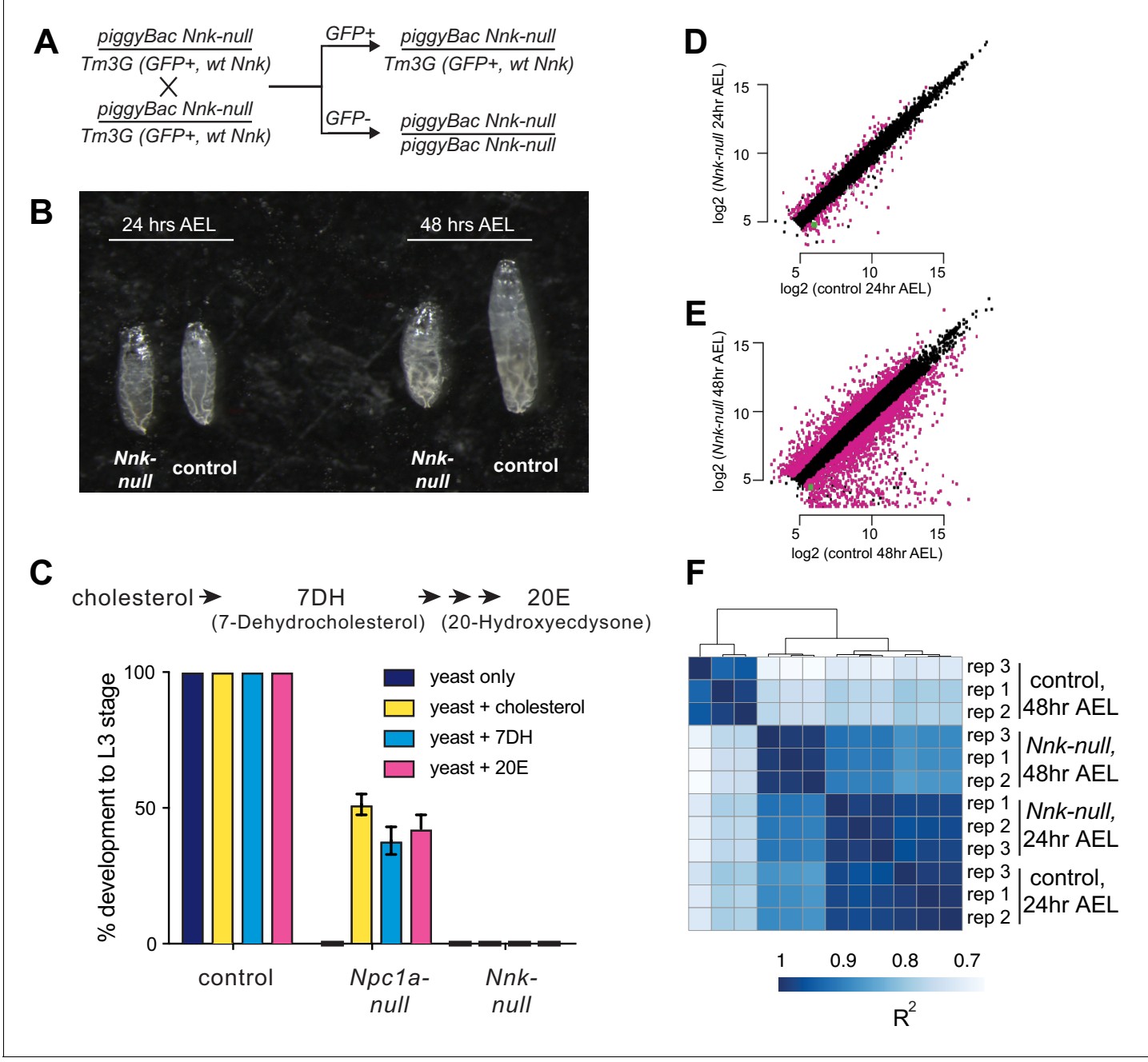

**Figure 4.** *Nicknack-null* larvae arrest in early larval development. (**A**) Schematic of cross to generate homozygous *piggyBac*-null larvae and age-matched control (heterozygote) siblings. *Tm3G* homozygous progeny are not shown, as they die as early embryos. (**B**) 24 hr after egg laying (AEL), *Nnk-null* larvae are morphologically indistinguishable from control larvae. 48 hr AEL, *Nnk-null* larvae are significantly smaller than the age-matched controls and have failed to undergo the first larval molt. (**C**) Adding sterols (cholesterol, 7DH or 20E) to the food of control (w1118) larvae did not alter their ability to develop to L3 larvae within 60 hr AEL. *Npc1a*-null larvae (*Npc1a57/Npc1a57*) used as a control do not develop on food supplemented with yeast alone but can be partially rescued (>40% molting into L3 larvae) with the addition of cholesterol, 7DH, or 20E. In contrast, *Nnk-null* mutants fail to progress through development even with the addition of dietary sterols. Vertical axis shows the percentage of L1 larvae that progressed to L3 larvae. Graphs show mean of three replicates with error bars showing standard deviation; n > 100 for all genotypes and treatments. (**D, E**) Scatter plots showing RNA-seq results for all expressed genes at 24 hr (**D**) and 48 hr (**E**) AEL. X-axis indicates the expression (normalized abundance) of genes in control larvae whereas the Y-axis shows the expression of genes in *Nnk-null* larvae at the same time point. Magenta dots represent genes significantly over- and under-expressed in *Nnk-null* larvae (green dot represents expression of *Nnk* itself). (**F**) Overall correlations between transcriptome profiles of three replicates each for control and *Nnk-null* larvae reveals that *Nnk-null* transcriptomes 48 hr AEL are more similar to transcriptomes of control and

*Figure 4 continued on next page*

*Figure 4 continued*

*Nnk-null* larvae 24 hr AEL than they are to age-matched controls, reflecting their developmental delay. Sample-to-sample Spearman $R^2$ distance matrix with hierarchical clustering using raw read counts of all 11,428 expressed genes.

significant over-representation of functional categories related to proteolysis and sterol transport (both important functions during larval development), as well as dopamine monooxygenase activity (*Supplementary file 3*). In contrast, we find that genes related to lysosome, cytochrome P450s (also important for larval molts), and eye-related functions are upregulated upon *Nnk* loss.

Second, we performed comparisons 48 hr AEL when the *Nnk-null* larvae are significantly smaller and appear developmentally arrested compared to their age-matched controls. At this timepoint, we find that 3027 (28.1% of all expressed genes) genes are differentially expressed, with 1301 genes at least two-fold upregulated and 1726 genes two-fold downregulated in the *Nnk-null* mutants compared to the age-matched controls (*Figure 4E*). Thus, there are significantly more genes affected by *Nnk* loss by 48 hr AEL. Intriguingly, clustering samples by the transcriptional profile of all genes shows that *Nnk-null* larvae at 48 hr AEL are transcriptionally more similar to control larvae at 24 hr AEL of either genotype than they are to age-matched control larvae (*Figure 4F*). The transcriptional status of *Nnk-null* larvae therefore mirrors the phenotypes we observe, displaying a severe developmentally arrested phenotype and transcriptional profile at 48 hr AEL (*Figure 4B*).

### *Nicknack* encodes a heterochromatin-localizing protein in *D. melanogaster*

Most of the *Odj-Nnk* cluster of *ZAD-ZNF* genes are functionally uncharacterized. Since *Odj* encodes a protein that is highly enriched in pericentric heterochromatin (*Swenson et al., 2016*), we speculated that its close paralog, *Nnk,* might also encode a protein with heterochromatic localization in *D. melanogaster* cells. To test this possibility, we used transient transfections to introduce epitope-tagged *Odj* and *Nnk* genes into *D. melanogaster* Schneider 2 (S2) cells and induced their expression with a heat-shock promoter (*Figure 5A*). Upon induction, we confirm that Oddjob has a broad localization pattern within heterochromatin (marked by histone H3 lysine nine methylation, H3K9me3 and outlined with a dashed line, Figure 5A). We found that the *Nnk*-encoded protein also localizes to heterochromatin, but its localization is restricted to discrete foci, unlike Oddjob (*Figure 5A*). Since heat-shock induction can alter chromatin properties in cells, we also employed a complementary transient transfection strategy in which we expressed mCherry epitope-tagged *Odj* and *Nnk* genes under the control of a constitutive pCopia promoter in S2 cells. These analyses also revealed a broad heterochromatic localization of Odj contrasting with discrete foci within heterochromatin for Nnk protein (*Figure 5B*). These foci do not overlap with centromeres (identified by the centromeric histone Cid) or dual-strand piRNA clusters (marked by the piRNA-binding HP1 protein Rhino) (*Figure 5—figure supplement 1*).

Odj has a broad heterochromatic localization in *D. melanogaster* cells, which could result from direct interaction with HP1a (*Swenson et al., 2016*). Indeed, Odj has two potential PxVxL motifs, which are putative interaction sites for HP1a (*Smothers and Henikoff, 2000*), in the linker and ZNF domains (*Figure 5—figure supplement 2A*). Investigating Odj orthologs in other *Drosophila* species revealed that the PxVxL motif in the linker domain is well-conserved, but the one in the ZNF domain is not. We found that mutation of the putative HP1a-interaction site in the linker domain (V164A) converted Odj localization from a broad to a discrete pattern that at least partially overlapped with Nnk (*Figure 5—figure supplement 2B*). In contrast, mutation of the second putative PxVxL site (V321A) did not significantly affect Odj localization. Our results suggest that the putative PxVxL motif in the Odj linker region is a major contributor to Oddjob's broad localization to heterochromatin potentially by mediating a direct interaction with HP1a. In the absence of this interaction, Oddjob and Nnk (which lacks a canonical PxVxL motif) localize similarly to discrete foci within heterochromatin (*Figure 5B*). Our findings suggest that altered protein–protein interactions or DNA-binding specificity via the linker domain may provide a means for functional diversification between closely-related *ZAD-ZNF* paralogs.

To gain deeper insight into the heterochromatic localization of Nnk and Odj, we performed chromatin immunoprecipitation and sequencing (ChIP-seq) using transient transfection of S2 cells with

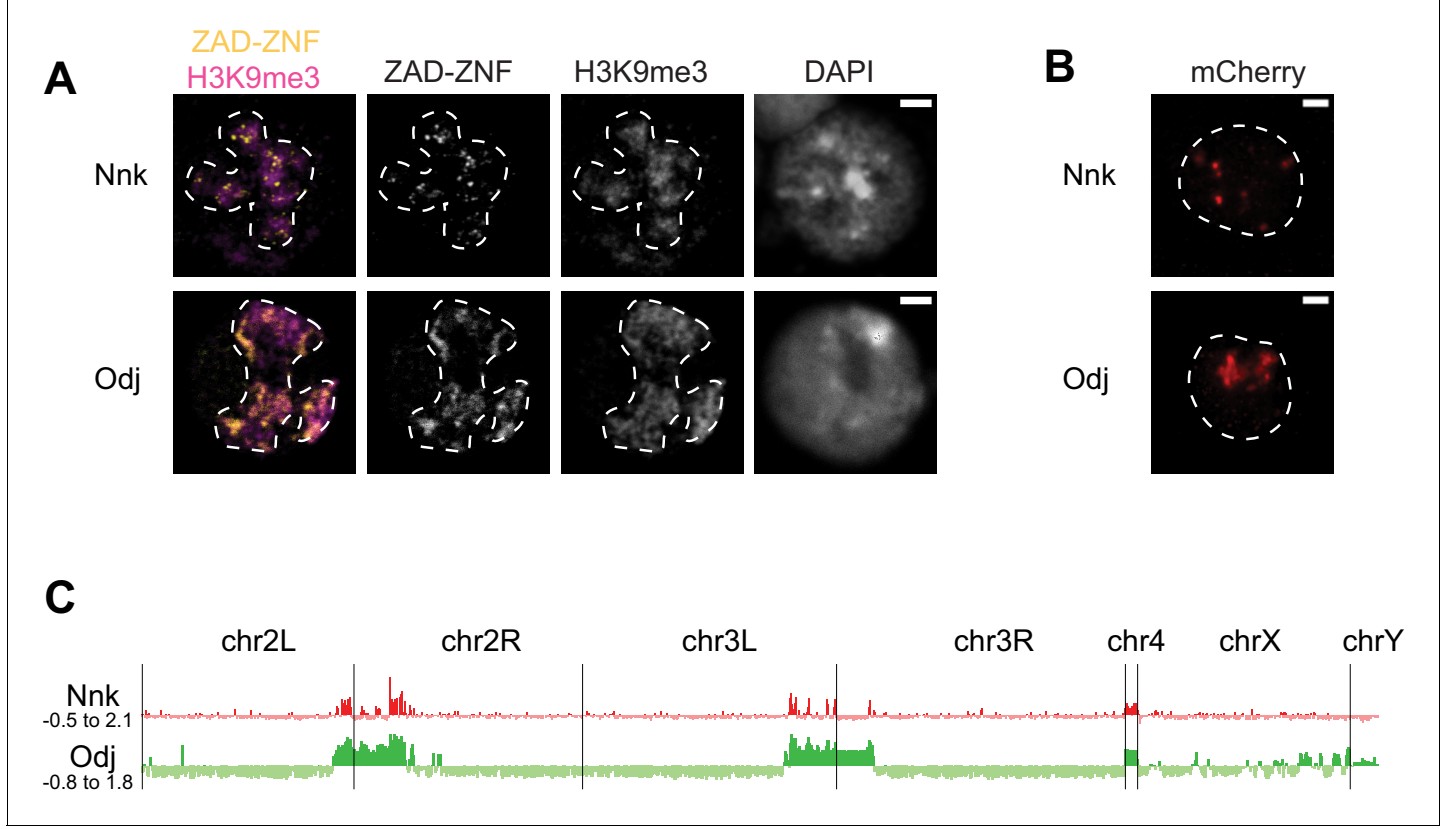

**Figure 5.** Heterochromatic localization of Odj and Nnk proteins. (**A**) Odj and Nnk proteins localize within the heterochromatic chromocenter, marked by H3K9me3 staining (magenta, outlined with dashed line) of *D. melanogaster* S2 cells. Whereas Oddjob localizes broadly to heterochromatin, Nicknack localizes to discrete foci within heterochromatin. All images are of representative nuclei from S2 cells transfected with a Venus-tagged ZAD-ZNF (yellow) under the control of an inducible heatshock-promoter. DAPI marks DNA in each nucleus. Scale bar = 2 μm. (**B**) Constitutive pCopia-driven expression confirms the broad Odj and discrete Nnk localization within heterochromatin of *D. melanogaster* S2 cells (dashed line outlines nucleus). Scale bar = 1 μm. (**C**) ChIP-seq analyses confirms that Odj binding is highly enriched throughout heterochromatin, whereas Nnk localizes only to some regions within heterochromatin, and to the 4[th] chromosome; note that S2 cells do not have an intact Y chromosome (*Lee et al., 2014*). The y-axis represents normalized ChIP-versus-input log2 ratios, using IGV to visualize and smooth data. Numbers on the y-axis give the range of ratios displayed for each factor.

The online version of this article includes the following figure supplement(s) for figure 5:

**Figure supplement 1.** Nnk localization in heterochromatin in *D. melanogaster* S2 cells.

**Figure supplement 2.** Mutation of the Odj PxVxL motif in the linker disrupts its broad localization within heterochromatin.

**Figure supplement 3.** Additional analysis of ChIP-seq signals.

**Figure supplement 4.** Effect of *Nnk* loss on gene and repeat expression.

the epitope-tagged pCopia constructs for *Odj* and *Nnk*. Consistent with its cytological localization (*Figure 5A and B*), we found that Odj was highly enriched throughout the large chromosome regions previously defined as pericentric heterochromatin (*Hoskins et al., 2015*) whereas Nnk-bound regions are more narrowly distributed within heterochromatin (*Figure 5C*). Finer-scale analyses suggested that Nnk and Odj signals are enriched close to transcription start sites classified by modENCODE as residing in TSS-proximal chromatin in S2 cells, both in heterochromatin and euchromatin (*Figure 5—figure supplement 3*; *Ho et al., 2014*). However, many of these regions overlap with false positive 'phantom peaks' previously identified in ChIP-seq experiments in S2 cells (*Jain et al., 2015*; *Figure 5—figure supplement 3*). While some heterochromatic TSSs could still represent true binding, we are more confident that the broader occupancy we observe outside TSSs represents Odj and Nnk's true localization. Moreover, we did not observe a significant effect of *Nnk* loss on heterochromatin-embedded gene expression at the larval L1 stage via RNA-seq analyses (*Figure 4D*, *Figure 5—figure supplement 4*). Instead, we found that some heterochromatin-

embedded repetitive elements are derepressed upon *Nnk* loss (*Figure 5—figure supplement 4*). Based on these results, we hypothesize that rather than acting as a transcription regulator for specific heterochromatin-embedded genes like its paralogs *Séance, Ouija Board* and *Molting Defective* (*Uryu et al., 2018*), *Nnk* may instead repress expression from heterochromatic repeats.

### *D. simulans Nnk* can rescue female, but not male viability of *Nnk-null D. melanogaster*

Our analyses revealed *Nnk* is an evolutionarily young, differentially-retained gene that is essential for larval development in *D. melanogaster. Nnk* has also evolved under dramatic positive selection in just the 2.5 million-year divergence between *D. melanogaster* and *D. simulans*, having undergone 52 fixed non-synonymous differences in the 439 aa protein-coding region (*Table 2*). Given the intriguing correlation between essentiality and positive selection that we had observed in the *ZAD-ZNF* genes, we investigated whether positive selection of *Nnk* has affected its function.

We first assayed the subcellular localization of epitope-tagged *Nnk* orthologs from *D. melanogaster* and *D. simulans* in *D. melanogaster* S2 cells. We used transient transfections to introduce these genes into S2 cells and used heat shock to drive their expression. We found that both proteins similarly localize to foci within heterochromatin (*Figure 6A–B*; dashed line marks heterochromatin boundaries). Thus, the rapid evolution of *Nnk* during *D. melanogaster- D. simulans* divergence has not dramatically affected its gross subcellular localization.

Next, we examined the consequences of *Nnk* positive selection on viability in *D. melanogaster*. We created a *D. simulans* Nnk 'rescue' transgene (*Figure 6C*). This rescue transgene is similar to the *D. melanogaster Nnk*-rescue construct (*Figure 3B*), except that it contains the *D. simulans Nnk* coding sequence (codon-optimized to *D. melanogaster*) with 1 kb *Nnk*-flanking sequences from *D. melanogaster* (*Supplementary file 4*). We introduced this *D. simulans Nnk*-rescue transgene into the same *attP* site on *D. melanogaster* X chromosome as the *D. melanogaster Nnk*-rescue transgene via PhiC31-mediated transgenesis (*Figure 6D*). The *attP*-insertion rescue design allowed us to put the transgene in the same genetic location in the *Nnk-mel* and *Nnk-sim* rescue crosses, normalizing for variability in expression of transgenes. This allowed a near-isogenic comparison of the *D. simulans* and *D. melanogaster Nnk* transgenes' ability to rescue inviability of *Nnk-null D. melanogaster* flies, despite their high level of sequence divergence. Unfortunately, low levels of endogenous *Nnk* expression did not allow us to assess whether the expression levels of both *Nnk* transgenes were equivalent.

We crossed heterozygous females either containing one or no copy of the (RNAi-resistant) *Nnk-rescue* transgene and the *Nnk-RNAi* allele to *Act5C-GAL4/CyO-GFP* males. We expected to find that progeny that did not inherit the rescue transgene would be inviable. In contrast, in the resulting progeny from rescue transgene-bearing females, we expect half of the progeny to inherit the *Nnk-rescue* transgene (see Materials and methods). We found that the *Nnk-mel* transgene significantly rescued *Nnk* knockdown compared to no-transgene controls; however, males were recovered at slightly lower levels than females (67 males: 101 females compared to expectation of 1:1 ratio; p=0.08, Fisher's exact test). We similarly found that the *D. simulans Nnk* transgene can significantly rescue the lethality caused by knockdown of endogenous *D. melanogaster Nnk* in females (*Figure 6C*), although at a slightly lower level than *Nnk-mel* rescue. In contrast, rescue of male viability by *D. simulans Nnk* is extremely poor (2 males: 33 females compared to expectation of 1:1 ratio; p<0.0001, Fisher's exact test) resulting in a severe sex-bias. Thus, *Nnk-sim* is much worse than *Nnk-mel* in rescuing male viability (67:101 versus 2:33, p<0.0001, Fisher's exact test). Based on these findings, we infer that the *D. simulans Nnk-rescue* transgene specifically fails to rescue *Nnk*-knockout males, in the presence of the heterochromatin-rich Y chromosome. Our findings suggest that not only is *Nnk* a positively-selected, essential *ZAD-ZNF* gene, but also that its positive selection is required for optimal function in the *D. melanogaster* genome.

## Discussion

In this study, we explored the relationship between genetic innovation and essentiality in the *ZAD-ZNF* gene family, which encodes the most abundant class of transcription factors in insects. Due to their lineage-specific amplification, protein structure and expression patterns, *ZAD-ZNF* genes were previously hypothesized to be analogous to the KZNF (KRAB-Zinc Finger) transcription factor-

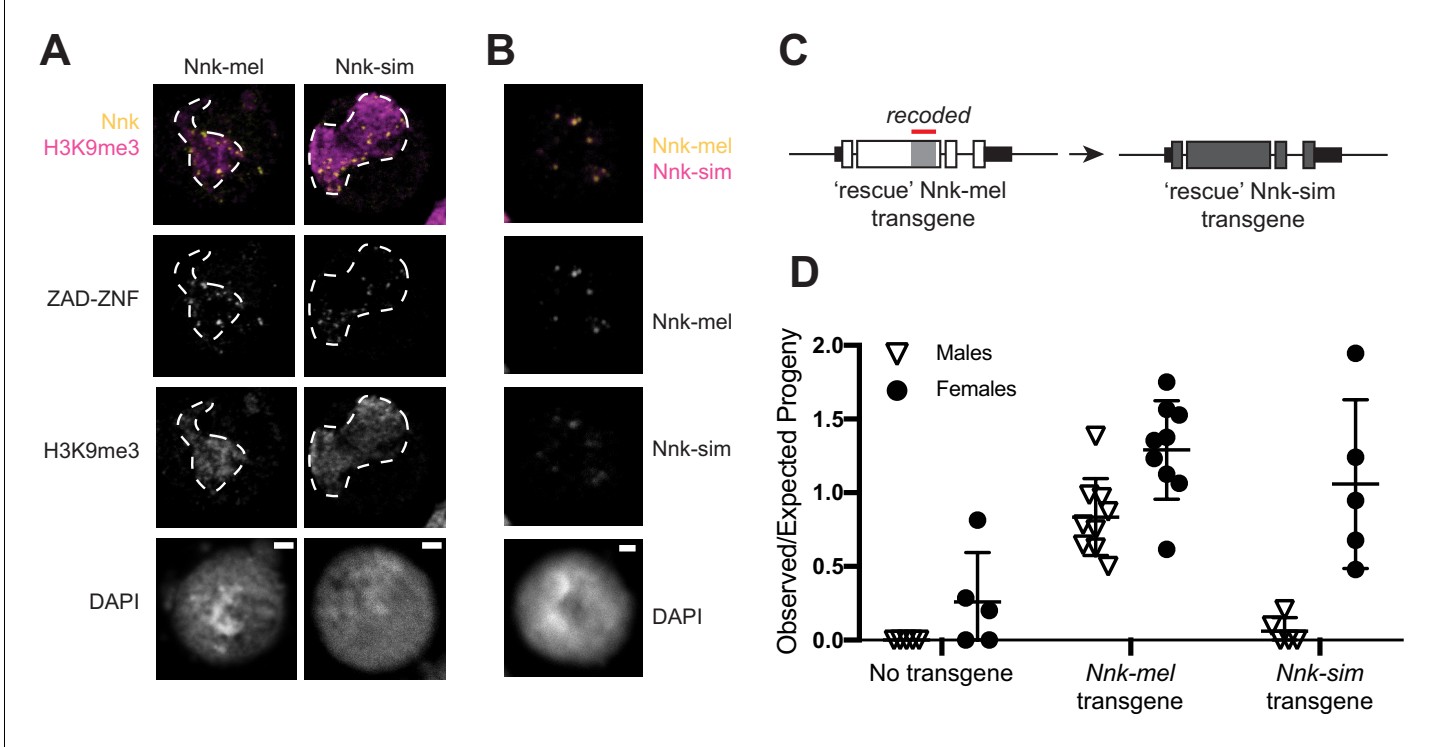

**Figure 6.** *D. simulans* Nicknack localizes to *D. melanogaster* heterochromatin but can only partially rescue essential function. (**A**) Venus-tagged *D. melanogaster* Nnk (Nnk-mel) or Venus-tagged *D. simulans* Nnk (Nnk-sim) (yellow) localize to foci within the heterochromatic chromocenter marked by H3K9me3 (magenta, dashed line) in *D. melanogaster* S2 cells. Each image is of a representative S2 cell nucleus with DAPI marking DNA. Scale bar = 2 µm. (**B**) Transient co-transfection of Venus-tagged Nnk-sim (magenta) and FLAG-tagged Nnk-mel (yellow) shows that they overlap in their localization to discrete heterochromatic foci in *D. melanogaster* S2 cells. Scale bar = 2 µm. (**C**) Schematic of rescue *Nnk-sim* transgene, which contains the genomic region 1 kb upstream of the *Nnk* start codon and 1 kb downstream of the stop codon, but with the *D. simulans* coding region (dark gray) from start to stop codon, including introns. The region of *Nnk-mel* targeted by the RNAi hairpin is highlighted by the red line; because of its divergence and codon-optimization, the *Nnk-sim* rescue transgene is resistant to RNAi. (**D**) Ability of *Nnk-mel* and *Nnk-sim* transgenes to rescue ubiquitous knockdown of *Nnk* using Act5C-GAL4 drivers. The vertical axis represents the observed/expected progeny in each cross. Without *Nnk* transgene, there are significantly fewer *Nnk* knockdown male and female progeny recovered than expected. *Nnk-mel* transgene is able to rescue *Nnk*-RNAi depletion in both males and females. *Nnk-sim* transgene is able to rescue *Nnk*-RNAi knockdown females significantly better than knockdown males. Filled circles represent number of female adult progeny and open triangles represent number of male adult progeny recovered from single-pair matings. Horizontal bars show the mean and standard deviation.

encoding gene family found in vertebrates (*Chung et al., 2002*; *Chung et al., 2007*), many of which target transposable element sequences (TEs) inserted in the genome. Just like *KZNF* genes in mammals, we find strong evidence for evolutionary dynamism in *Drosophila ZAD-ZNF* genes. For example, we find that only 61 of 91 *ZAD-ZNFs* found in *D. melanogaster* are universally retained in most *Drosophila* species. Furthermore, 12 *ZAD-ZNFs* have evolved under positive selection during *D. melanogaster*- *D. simulans* divergence.

Despite these similarities, however, there are considerable differences between these gene families. First, a direct connection to TEs has only been revealed for one *ZAD-ZNF* gene in *Drosophila*; *CG17801* has a role in regulating *HetA* and *Blood* transposable elements in the female germline (*Czech et al., 2013*). Second, we find that *ZAD-ZNF* genes that evolve under positive selection are more likely to encode essential functions in embryonic axial patterning, larval development, and meiosis (*Chen et al., 2000*; *Harms et al., 2000*; *Page and Orr-Weaver, 1996*; *Uryu et al., 2018*; *Lake et al., 2011*; *Chen et al., 2006*; *Table 2*). In contrast, most *KZNF* genes that have been shown to be essential for sterility or viability in mammals are slowly evolving (*Wolf et al., 2020*; *Imbeault et al., 2017*). Finally, unlike *KZNFs* (*Thomas and Schneider, 2011*), positive selection in *ZAD-ZNFs* is not primarily focused on their DNA-binding C2H2 domains but rather on the poorly-characterized linker domains that connect the ZAD and C2H2 domains (*Table 2*). We speculate that

the linker, which is often comprised of intrinsically disordered domains, may play an important role in chromatin localization of ZAD-ZNF proteins either via direct DNA-binding or via protein–protein interactions (*Brodsky et al., 2020*; *Erijman et al., 2020*). For example, the Oddjob linker domain encodes a PxVxL motif that is important for its broad heterochromatic localization, potentially by mediating direct interaction with HP1a (*Figure 5*, *Figure 5—figure supplement 2*).

Based on these dissimilarities, we do not favor the possibility that ZAD-ZNF innovation is solely driven by arms-races with TEs. Instead, we favor the hypothesis that the recurrent adaptation of a subset of *ZAD-ZNF* stems from their roles in pericentromeric heterochromatin organization or regulation of gene expression. Indeed, many genes that encode crucial heterochromatin functions are often critical for viability or fertility, yet are quite variable even among closely-related species (*Ross et al., 2013*; *Klattenhoff et al., 2009*; *Levine et al., 2016*; *Parhad et al., 2017*; *Vermaak and Malik, 2009*). Heterochromatin is a gene-poor component of most eukaryotic genomes. Yet, its establishment and maintenance is nevertheless essential for many cellular processes including chromosome condensation and segregation, repression of TEs, and genome stability (*Abe et al., 2016*; *Grézy et al., 2016*; *Levine et al., 2015*; *Nambiar and Smith, 2018*; *Okita et al., 2019*; *Vernì and Cenci, 2015*; *Liu et al., 2014*; *Azzaz et al., 2014*; *Inoue et al., 2008*; *Ruiz-Estévez et al., 2014*; *Verschure et al., 2005*; *Brennecke et al., 2007*; *Senti and Brennecke, 2010*; *Goriaux et al., 2014a*; *Goriaux et al., 2014b*). Thus, the rapid evolution of genes encoding heterochromatin functions might reflect lineage-specific mechanisms to package heterochromatic DNA, or silence TEs. In this context, it is important to note the heterochromatic satellite DNA sequences themselves are among the most rapidly evolving component of *Drosophila* genomes (*Wei et al., 2018*; *Chakraborty et al., 2020*; *Jagannathan et al., 2017*).

Several characterized *ZAD-ZNFs* have been found to play key roles at heterochromatin. For example, the Oddjob protein co-immunoprecipitates with HP1a and broadly localizes to heterochromatin (*Figure 5*; *Swenson et al., 2016*). Similarly, Séance and Ouija Board control the expression of heterochromatin-embedded genes necessary for larval development (*Uryu et al., 2018*). Based on these observations, we propose that *ZAD-ZNF* diversification (marked by gene turnover and positive selection) is driven by the high turnover of sequences embedded within heterochromatin. Although the bulk of heterochromatin is made up of highly repetitive elements such as satellite DNAs and TEs, heterochromatin also harbors many genes that are deeply embedded within heterochromatin (*Yasuhara et al., 2005*; *Yasuhara and Wakimoto, 2006*; *Eberl et al., 1993*; *Schulze et al., 2006*; *Schulze et al., 2005*; *Devlin et al., 1990*). These genes, many of which encode essential functions (*Sinclair et al., 2000*), require a heterochromatic environment to ensure their correct expression and regulation (*Wakimoto and Hearn, 1990*). We posit that the constant turnover of flanking and embedded sequence elements such as TEs and satellite DNAs may require constant adaptation of transcription factors required for the proper expression of genes embedded in heterochromatin. *ZAD-ZNF* adaptation might also be necessary to protect against the inappropriate expression of heterochromatin-embedded elements especially at crucial developmental stages. Because of the high rate of evolutionary turnover of heterochromatic sequences, we hypothesize that *ZAD-ZNF* genes that are essential in one species could nevertheless be lost in another, either because the target loci of the *ZAD-ZNF* genes is lost, or because another *ZAD-ZNF* paralog has acquired this essential regulation function.

By testing the causal link between positive selection and essential function, we also find further support for *Nnk*'s essential function being related to heterochromatin biology. We find that *D. simulans Nnk* can significantly rescue *Nnk-null* inviability in females, but not in males. We speculate that this failure to rescue male viability could be due to the heterochromatin-rich Y chromosome in males. For example, if the *D. melanogaster* Nnk protein, but not the *D. simulans* Nnk protein, could appropriately repress the *D. melanogaster* Y chromosome, this might explain the sex-bias seen in the *D. simulans Nnk*-rescue progeny. The Y chromosome is itself not essential for viability or sex determination in *Drosophila*. However, de-repression of the heterochromatin-rich Y chromosome could nevertheless lead to detrimental consequences on larval development. This effect could be direct, leading to inappropriate expression of Y-chromosome-embedded genetic elements that block larval development. Alternatively, this effect could be indirect; de-repression of the Y chromosome could indirectly impact several other chromatin processes genome-wide e.g., due to inappropriate titration of transcription or heterochromatin factors, which could exacerbate an already hypomorphic function of the *D. simulans Nnk* allele (*Branco et al., 1869*; *Francisco and Lemos, 2014*; *Wang et al., 2018*;

*Piergentili, 2010*; *Brown et al., 2020*). Finally, different functional optimality of male-specific or female-specific *Nnk* functions could also drive its rapid evolution (*VanKuren and Long, 2018*). In any of these scenarios, we hypothesize that rapid co-evolution with heterochromatic sequences might have driven the rapid evolution of Nnk and possibly other heterochromatin-interacting ZAD-ZNF proteins.

Based on our findings, we hypothesize that constant adaptation in *ZAD-ZNF* genes is driven by rapid alterations in heterochromatin across *Drosophila* and other insect species. This co-evolutionary arms-race may provide the explanation for the unexpected correlation we find between gene essentiality and innovation in the largest family of transcription factors in insect genomes.

## Materials and methods

### Phylogenomic analysis of *ZAD-ZNF* genes in *Drosophila*

We used the Flybase database (http://flybase.org) to identify all ZAD-containing proteins (Pfam motif PF07776) in 12 sequenced and annotated *Drosophila* species (*Drosophila 12 Genomes Consortium, 2007*). Using NCBI's Conserved Domains search, we identified other domains found in the ZAD-containing proteins (*Marchler-Bauer et al., 2017*). To estimate the evolutionary age of these ZAD-containing genes, we used OrthoDB to identify orthologs of the 91 ZAD-containing proteins across *Drosophila* (*Zdobnov et al., 2017*). We used the divergence between the two most distantly-related species that still encoded and ortholog to calculate the minimum age of each gene using timetree.org (*Kumar et al., 2017*). If orthologs were identified in basally-branching but not later-branching species by OrthoDB, we defined this as a loss of the *ZAD-ZNF* gene in the lineage. The repertoire of different *ZAD-ZNF* genes in different *Drosophila* species inferred from our analysis is summarized in *Supplementary file 1*.

### Defining gene essentiality

We used FlyBase gene summaries (FlyBase.org) and published studies when available to define gene essentiality. Our criteria for essentiality were broad: if there was a lethal allele reported in any assay, we counted the gene as essential for viability in *D. melanogaster*. *Supplementary file 1* summarizes the phenotypic data (and its source) that we used to classify the *ZAD-ZNF* genes into either essential or non-essential categories.

### Analyses of positive selection

For the McDonald-Kreitman test, we extracted the gene of interest from *D. melanogaster* population genetic datasets available through Popfly (www.popfly.com) and removed low frequency (<0.05) variants from the dataset to minimize the effects of false positives and low-frequency variants that may not have been subject to selection (*Hervas et al., 2017*; *Fay et al., 2002*). We used a manually trimmed alignment of the *D. melanogaster* filtered dataset and the reference *D. simulans* sequence for the McDonald-Kreitman test (http://mkt.uab.es/mkt/; *McDonald and Kreitman, 1991*; *Egea et al., 2008*). We also carried out a polarized McDonald-Kreitman test in which we only analyzed non-synonymous and synonymous sites that were fixed along the lineage leading to *D. melanogaster*, after its divergence from *D. simulans*, inferred using *D. yakuba* as an outgroup species, except for the *Trem* gene where we used *D. erecta* outgroup instead because the *D. yakuba* ortholog aligned poorly to *D. melanogaster* and *D. simulans*.

### Viability studies

We used an *Act-GAL4/CyO-GFP* driver for ubiquitous knockdown. The RNAi lines used to specifically target *Oddjob* cluster genes are the VDRC KK or GD lines: *Oddjob* (27971), *CG17803* (38869), *CG17801* (29501), *CG17806* (40106) and *Nicknack* (102311). RNAi controls used for the experiment were *Cid* (43856) and *HP1B* (26097) for the ubiquitous knockdown. Ubiquitous knockdown of *Cid* produced no viable progeny. We crossed five virgin females carrying *Act-GAL4/CyO-GFP* to 3 males of each RNAi line. We allowed the females to lay eggs for 3 days and flipped the flies into fresh vials three times. Each cross was performed in triplicate. Progeny were counted 10–15 days after each cross was set up. A minimum of 20 *CyO-GFP* males and 20 *CyO-GFP* females (*i.e.,* control genotype) were required for us to quantify the crosses. Each cross is represented by a point on the graph and

shown as a ratio of observed (number of knockdown or rescue progeny counted in each cross) over expected (the number of expected knockdown or rescue progeny from Mendelian segregation inferred by inheritance of the balancer chromosome). If the knockdown has no effect on viability or the rescue is 100% effective, the observed/expected = 1. If the knockdown of the essential gene is 100% penetrant or the rescue is ineffective, the observed/expected = 0. Plotting and statistical analyses were conducted using Graphpad Prism 8 software.

## Defining orthologs

Since *Oddjob* cluster genes experienced numerous independent segmental duplications, it was not possible to determine orthologs by synteny alone. Instead, we used TBLASTN (*Gertz et al., 2006*) to identify candidate orthologs of *Oddjob* cluster genes, using the genes in *D. melanogaster* as queries. We used a reciprocal blast search strategy to identify potential orthologs and further investigated these candidates by making a maximum likelihood phylogenetic tree (LG substitution model in PhyML with 100 bootstrap replicates) of a manually trimmed protein alignment (constructed using Clustal Omega program [*Thompson et al., 2002*] in the Geneious package, www.geneious.com) (*Supplementary file 2*, *Figure 2—figure supplement 1*). We assigned orthologs based on genes that formed a monophyletic clade with the each of the *D. melanogaster Oddjob* cluster genes. We mapped these *Oddjob* cluster orthologs back to each genome assembly to determine the composition of the *Oddjob* locus across *Drosophila* species. In cases where there were other ZAD-ZNFs present, we blasted them against the *D. melanogaster* genome to examine if there were any orthologs present in *D. melanogaster.* If the top hit was not a member of the *Oddjob* locus, we did not include it in the tree. In the case where there are partial ZAD-ZNFs (containing just the ZAD domain or the zinc-finger domains), we performed a blast search against the *D. melanogaster* genome. If the top hit was a member of the *Oddjob* cluster, we assigned orthology by making a phylogenetic tree with all other *Oddjob* cluster orthologs. Our phylogenomic inferences are dependent on the accuracy and state of completion of genomes from other *Drosophila* species, which can vary.

## *Nnk*-rescue transgene design, construction, and crosses

We designed a *D. melanogaster* recoded transgene comprising a 3.3 kb fragment containing the genomic region of *Nicknack* plus 1 kb upstream and downstream of the start and stop codons, based on the *D. melanogaster* reference assembly. We recoded the sequence targeted by the VDRC RNAi KK line (103211) by making synonymous changes at each codon. The synonymous changes made the gene resistant to RNAi knockdown but did not alter the amino-acid sequence of Nnk. The recoded region spans 270 base pairs that corresponds to the region targeted by the VDRC RNAi KK line (103211). The resulting sequence was synthesized by GENEWIZ Co. Ltd. (Suzhou, China) and cloned into a plasmid we generated that contains 3xP3-DsRed attP, which produces fluorescent red eyes in the adult to mark the presence of the transgene. To generate the *D. simulans* transgenic allele, we codon-optimized the *D. simulans Nnk* coding sequence for the *D. melanogaster* genome using IDT's codon-optimization tool. The resulting sequence was synthesized by GENEWIZ Co. Ltd. (Suzhou, China) and swapped for the *D. melanogaster Nnk* coding sequence in the plasmid described above, using the NEBuilder kit (New England Biolabs). We submitted transgenic constructs to The BestGene Inc (Chino Hills, CA) for injection into the X-chromosome *attP18* line (BL 32107) using *PhiC31* site-specific integration (*Groth et al., 2004*).

For the transgene rescue cross, we crossed five virgin female flies bearing one copy of *Nnk-rescue* transgene (on the X chromosome) and the RNAi allele (on the second chromosome) to 3 *Act5C-GAL4/CyO-GFP* males. We allowed the females to lay eggs for 3 days and flipped the cross three times. We set up each cross in triplicate and progeny were counted after 10–15 days. The description of the observed/expected calculation can be found in 'viability studies' section. Each cross had at least three replicates. All flies were raised at 25˚C.

## Characterization of *Nnk-null* mutants

We placed 50–75 flies heterozygous for the *Nnk-null* allele (*Nnk pBac null/TM3G)* or for the *Nnk* CRISPR allele (*Nnk CRISPR/TM3G)* into a small embryo collection cage containing a grape-juice plate with a thin strip of yeast paste and collected embryos for 3 hr at 25˚C. We transferred the larvae to fresh grape-juice plates containing yeast paste daily and scored developmental stage by mouth

hook morphology. We used fluorescence to distinguish between heterozygotes (GFP-positive larvae) and homozygotes (GFP-negative larvae). For the trans-heterozygote evaluation, we crossed 30–40 virgin female *Nnk CRISPR/TM3G* to 10 *Nnk pBac null/TM3G* males. Crosses were done in triplicate and at least 100 progeny were counted per cross.

## Larval collection for RNA-sequencing

We placed 50–75 flies heterozygous for the *Nnk-null* allele (*Nnk pBac/TM3G)* into a small embryo collection cage containing a grape-juice plate with a thin strip of yeast paste and collected embryos for 3 hr at 25°C. The first time point was collected 24 hr after egg laying (AEL) and the second 48 AEL. We transferred the larvae to fresh grape-juice plates containing yeast paste daily.

## RNA extractions and library preparation

Whole larvae (~30 animals at 24 hr AEL and ~20 animals at 48 hr AEL for each sample; RNA from each time point and genotype was prepared in triplicate) were ground in a 1.5 mL Eppendorf tube containing 50 µL of TRIzol reagent using a DNase, RNase and DNA free 1.5 mL pestle. 450 µL of TRIzol reagent was added after grinding. Immediately, we added 500 µL of chloroform and the tube was inverted gently 2–3 times. We removed the aqueous phase into a fresh tube containing 1 mL of 200 proof EtOH and mixed by inversion. The mixture was then bound to a Zymo-spin column according to the manufacturer's instructions (Zymo Research). We followed the DNase extraction and purification protocol outlined in the RNA Clean and Concentrator kit (Zymo Research). We eluted the RNA in 15 µL of DNase/RNase-free water and immediately placed the samples at −80°C. We checked the quality of the samples with a 2200 Tapestation (Agilent Technologies) and selected samples that had an RNA integrity number >9.0 for library preparation. Library construction and Illumina 150 bp paired-end RNA-sequencing were conducted at Novogene Bioinformatics Technology Co., Ltd (Beijing, China).

## Transcriptome data analysis

We used Kallisto (*Bray et al., 2016*) to quantify abundances of the *D. melanogaster* reference transcriptome (refMrna.fa for dm6, obtained from UCSC Genome Browser Oct 16[th], 2018, which contains 34,114 transcripts). For each transcript, we acquired the gene name using R (org.Dm.eg.db). Kallisto counts were read into R using the tximport package (*Soneson et al., 2015*) and using the summarizeToGene function, we summarized alternative splice-form counts into a single count per gene. We used DESeq2 to identify differentially expressed genes with adjusted p-value<=0.05 and absolute $\log_2$(fold change)>=1 (*Anders and Huber, 2010*), comparing *Nnk-null* larvae to controls for each timepoint separately. Before performing each comparison, we excluded genes with low expression (<100 counts total across all samples): this filtering yields 10,574 'expressed' genes in 24 hr AEL larvae, and 10,758 'expressed' genes in 48 hr AEL larvae. The RNA-seq data has been deposited in the SRA database with the accession number PRJNA643855.

We performed Gene Ontology (GO) enrichment analysis on each of four gene lists: 116 over-expressed genes and 133 under-expressed genes in 24 hr AEL larvae, and 1301 over-expressed genes and 1726 under-expressed genes in 48 hr AEL larvae. We used the Bioconductor GOstats package to perform conditional hypergeometric enrichment tests for each of three ontologies (biological process, molecular function and cellular component) (*Huber et al., 2015*; *Falcon and Gentleman, 2007*). For the 'universe' of all genes examined (*i.e.,* background) we used the corresponding list of 'expressed' genes at each developmental timepoint. We report only over-represented categories with p<0.001, and use annotations found in the org.Dm.eg.db Bioconductor package.

To estimate expression levels aggregated across all instances of each repeat type, we took an approach similar to that of *Day et al., 2010*. Here, rather than mapping reads to the typical reference genome assembly, we constructed a 'repeat assembly', where we used RepeatMasker annotations for dm6 (obtained from UCSC) to extract and concatenate all instances of each repeat type, adding 75 bp flanking sequences (half the length of each read) and inserting 150 N bases between each instance. The repeat assembly therefore consists of a single 'chromosome' for each repeat type. We then used BWA-mem to map sequences as single reads (not paired-ends) to the repeat assembly (*Li, 2013*), and samtools to count reads mapping to each repeat type ('chromosome'). We combined counts for pairs of simple repeats that represent the reverse-complement of one another.

We used DESeq2 to perform statistical analyses on raw counts, using the total number of sequenced fragments as size factors (*Love et al., 2014*). We normalized counts by dividing each by the number of reads sequenced for that sample in millions.

## Dietary sterol supplementation

To evaluate the ability of dietary sterols to rescue *Nnk-null* phenotype, we carried out sterol supplementation as previously outlined (*Uryu et al., 2018*). Briefly, we mixed together 20 mg of dry yeast in 38 µL of water. To this yeast paste, we added either 2 µL of EtOH (negative control) or 2 µL of EtOH plus cholesterol (Sigma), 7-dehydrocholestrol (Sigma) or 20-hydroxyecdysone. Stocks used for these experiments were balanced over GFP-balancers (*Npc1a57/CyO-GFP* and *Nnk-null/TM3G*). The control used for these experiments was the *w1118* stock. Eggs were laid on yeasted grape-juice plates for 3 hr at 25°C. 24 hr AEL, GFP-negative larvae were transferred onto grape plates containing fresh yeast paste at 25°C. For 72 hr AEL, larvae were transferred to fresh grape-juice plates containing yeast paste daily and scored for their developmental stage based on the morphology of their mouth hooks. *Npc1a57/CyO-GFP* stocks were a kind gift from Leo Pallanck (*Fluegel et al., 2006*).

## Tissue culture and transfection

*Oddjob* cluster genes were amplified from genomic DNA from *D. melanogaster* and *D. simulans* and directionally cloned into pENTER/D-TOPO (ThermoFisher) according to the manufacturer's instructions. We verified that clones had the appropriate insertions by sequencing. We used LR clonase II (ThermoFisher) to get each *Oddjob* cluster gene into the *Drosophila* Gateway Vector destination vector to express the gene of interest with an N-terminal Venus tag under the control of the *D. melanogaster Hsp70* promoter (pHVW). For Rhino localization in S2 cells, the *Rhino* gene was cloned into the *Drosophila* Gateway Vector destination vector to enable expression of Rhino with an N-terminal 3xFlag tag under the control of the *D. melanogaster Hsp70* promoter (pHFW). *Nnk* and *Odj* were also cloned into pCopia vectors containing N-terminal and C-terminal mCherry epitope tags, respectively, and HP1a was previously tagged with mCerulean at the N-terminus.

Schneider 2 cells were obtained from the *Drosophila* Genomics Resource Center (Bloomington, IN, USA) and grown at 25°C in M3+BPYE+10%FCS. For the transfections, one million cells were seeded, and one day later 2 micrograms of plasmid DNA was transfected into cells using Xtremegene HP transfection reagent (Roche) according to the manufacturer's specifications. For the Rhino localization experiment, cells were co-transfected with 1 µg of GFP-tagged ZAD-ZNF vector and 1 µg of Flag-tagged Rhino vector. Cells were allowed to recover for 24 hr post transfection, heat shocked for 1 hr at 37°C and recovered for 3 hr at 25°C prior to fixation. For the pCopia vectors, transient transfections were conducted on S2 cells using the TransIT-2020 reagent (Mirus), and live imaging was performed 72 hr later, using an Applied Precision Deltavision microscope and analyzed using SoftworX software.

For the heatshock vector transfections, cells were transferred to coverslips for 30–45 min prior to starting the immunohistochemistry protocol. 0.5% sodium citrate hypotonic solution was added to the coverslip for 10 min to swell cells, which was then spun at 1900 rpm for 1 min in a Cytospin III to remove the cytoplasm from the cells. The sodium citrate was immediately removed and cells were subsequently fixed. For fixation, 4% PFA + PBST (PBS + 0.1% Triton) was added to the cells for 10 min. Coverslips were then washed in PBST and blocked for 30 min in PBST + 3% BSA. Cells were incubated with primary antibody overnight at 4°C in a humid chamber. For immunolocalization, the following dilutions of primary antibodies were used: GFP (Abcam AB13970) 1:1000, H3K9me3 (Abcam AB8898) 1:500 M2 FLAG (Sigma-Aldrich F4042) 1:1000. After washing with PBST three times for 10 min per wash, the following fluorescent secondary antibodies were used at 1:1000 dilution: goat anti-chicken (Invitrogen Alexa Fluor 488, A-11039), goat anti-rabbit (Invitrogen Alexa Fluor 568, A-11011) and goat anti-mouse (Invitrogen Alexa Fluor 568, A-11031). The cells were incubated with 1x DAPI in the final wash and mounted in SlowFade Gold Mounting Medium (ThermoFisher). We imaged cells on a Leica TCS SP5 II confocal microscope with LASAF software and images were processed using ImageJ and were representative of the cell population.

## ChIP-seq analyses

S2 cells transfected with either mCherry-Nnk or Odj-mCherry for 72 hr were fixed with 1% paraformaldehyde for 10 min and sheared with Bioruptor sonicator (Diagenode) to obtain chromatin. Chromatin fragments were confirmed to contain DNA in the 200–500 bp size range using a Bioanalyzer (Agilent). Each immunoprecipitation was performed on chromatin from $2 \times 10^7$ S2 cells by overnight incubation with Protein-G Dynabeads and 5 µg of anti-mCherry antibody (Novus). Library construction from immunoprecipitated DNA was conducted using TruSeq sample preparation kits (Illumina). 150 bp paired-end sequences were generated by the Vincent J. Coates Genomics Sequencing Laboratory at UC Berkeley.

We used BWA-mem to map paired reads to version 6 of the *D. melanogaster* genome assembly (dm6) from which we had removed unplaced scaffolds (*Li, 2013*). We used deepTools' bamCompare (with the '–binsize 1 --extendReads' options) to obtain log2 ratios of fragment coverage for matched ChIP and input samples, and visualized those ratios in IGV (*Robinson et al., 2011*). We further visualized ChIP-seq signal around TSSs using deepTools' computeMatrix and plotHeatmap tools, using TSS annotations obtained from the TxDb.Dmelanogaster.UCSC.dm6.ensGene BioConductor package, taking the most upstream TSS for genes with alternative start sites. We split TSS annotations according to whether they were within 'TSS-proximal' (active) chromatin according to modENCODE's nine state annotation for S2 cells, obtained from http://intermine.modencode.org and converted from dm3 to dm6 coordinates using UCSC's liftOver tool (*Hinrichs et al., 2006*). We further split the 'active' TSSs according to whether they are within cytogenetic heterochromatin, using coordinates from *Supplementary file 2* of *Hoskins et al., 2015* as well as the whole of chromosomes four and Y. The ChIP-seq data has been deposited in the SRA database with the accession number PRJNA644950.

## Acknowledgements

We thank past and present members of the Malik lab for their comments and discussion. In addition, we are especially grateful to Celeste Berg and Barbara Wakimoto for their helpful comments during this project, and to Leo Pallanck for helpful discussions and generous gift of fly strains. We also thank Eric Lai for sharing the CRISPR-null *Nnk* fly line. We thank Ching-Ho Chang, Michelle Hays, Pravrutha Raman, and Aida de la Cruz for their comments on the manuscript. This work was supported by the following grants: NIH training grant T32 CA009657, NIH F30 CA225077 fellowship, and Julie Tall Achievement Rewards for College Scientists (ARCS) endowment from the Seattle Chapter of the ARCS Foundation (to BK); NIH R01 GM117420 (to GHK), NIH R01 GM074108 and HHMI Investigator grant (to HSM). The funders had no role in study design, data collection and analysis, decision to publish, or preparation of the manuscript. The authors declare that they have no conflicts of interest.

## Additional information

### Funding

| Funder | Grant reference number | Author |
| --- | --- | --- |
| National Cancer Institute | T32 CA009657 | Bhavatharini Kasinathan |
| National Cancer Institute | F30 CA225077 | Bhavatharini Kasinathan |
| Achievement Rewards for College Scientists Foundation | | Bhavatharini Kasinathan |
| National Institute of General Medical Sciences | R01 GM117420 | Gary H Karpen |
| National Institute of General Medical Sciences | R01 GM074108 | Harmit S Malik |
| Howard Hughes Medical Institute | Investigator award | Harmit S Malik |

The funders had no role in study design, data collection and interpretation, or the decision to submit the work for publication.

## Author contributions
Bhavatharini Kasinathan, Harmit S Malik, Conceptualization, Data curation, Formal analysis, Supervision, Funding acquisition, Validation, Investigation, Visualization, Methodology, Writing - original draft, Project administration, Writing - review and editing; Serafin U Colmenares III, Data curation, Formal analysis, Validation, Investigation, Visualization, Methodology, Writing - review and editing; Hannah McConnell, Validation, Investigation, Methodology, Writing - review and editing; Janet M Young, Resources, Data curation, Software, Supervision, Investigation, Methodology, Writing - original draft, Writing - review and editing; Gary H Karpen, Supervision, Funding acquisition, Validation, Writing - original draft, Project administration, Writing - review and editing

## Author ORCIDs
Bhavatharini Kasinathan (iD) https://orcid.org/0000-0001-7053-5826
Serafin U Colmenares III (iD) http://orcid.org/0000-0002-4094-4220
Janet M Young (iD) https://orcid.org/0000-0001-8220-8427
Gary H Karpen (iD) http://orcid.org/0000-0003-1534-0385
Harmit S Malik (iD) https://orcid.org/0000-0001-6005-0016

## Decision letter and Author response
Decision letter https://doi.org/10.7554/eLife.63368.sa1
Author response https://doi.org/10.7554/eLife.63368.sa2

# Additional files

## Supplementary files
• Supplementary file 1. Differential retention in *Drosophila* species and phenotypes conferred by the 91 *ZAD-ZNF* genes in *D. melanogaster*. ZAD-ZNF gene repertoires in different *Drosophila* species were inferred from OrthoDB analyses (Materials and methods). Phenotypic consequences of knockdown or knockout phenotype of ZAD-ZNFs in *D. melanogaster* were retrieved from flybase.org. Key for citations: Neely et al. Cell 2010 (*Neely et al., 2010*) Mummery-Widmer et al. Nature 2009 (*Mummery-Widmer et al., 2009*) Schnorrer et al. Nature 2010 (*Schnorrer et al., 2010*) Chang et al. Dev. Cell 2010 (*Chang et al., 2010*) Burguete et al. Dev. Biol. 2019 (*Burguete et al., 2019*) Gibert et al. Dev. Biol. 2005 (*Gibert et al., 2005*) Dai et al. Dev. Cell 2003 (*Dai et al., 2003*) Komura-Kawa et al. PLOS Genet. 2015 (*Komura-Kawa et al., 2015*) Weiler Genetics 2007 (*Weiler, 2007*) Giulian et al. PLOS ONE 2013 (*Giuliani et al., 2013*) Chen et al. Science 2010 (*Chen et al., 2010*) Schalet Mutat. Res. 1986 (*Schalet, 1986*) Shannon et al. Genetics 1972 (*Shannon et al., 1972*) Judd et al. Genetics 1972 (*Judd et al., 1972*) Liu et al. Genetics 1975 (*Liu and Lim, 1975*) Lim et al. Genet. Res. 1974 (*Lim and Snyder, 1974*) Robbins Genetics 1983 (*Robbins, 1983*) Gaszner et al. Genes Dev. 1999 (*Gaszner et al., 1999*) Schertel et al. Genome Res. 2015 (*Schertel et al., 2015*) Toba et al. Genetics 1999 (*Toba et al., 1999*) Ruppert et al. Cell Rep. 2017 (*Ruppert et al., 2017*) Schupbach et al. Genetics 1989 (*Schüpbach and Wieschaus, 1989*) Chen et al. Development 2000 (*Chen et al., 2000*) Lieberfarb et al. Development 1996 (*Lieberfarb et al., 1996*) Page et al. J. Cell Sci. 1996 (*Page and Orr-Weaver, 1996*) Whitfield et al. PLOS Biol. 2013 (*Whitfield et al., 2013*) Holden et al. Genetics 1973 (*Holden and Suzuki, 1973*) Ishimoto et al. Genetics 2010 (*Ishimoto and Kitamoto, 2010*) Ishimoto et al. Proc. Natl. Acad. Sci. USA 2009 (*Ishimoto et al., 2009*) Walker et al. J. Insect Physiol. 1987 (*Walker et al., 1987*) Ono et al. Dev. Biol. 2006 (*Ono et al., 2006*) Neubueser et al. Dev. Biol. 2005 (*Neubueser et al., 2005*) Laundrie et al. Genetics 2003 (*Laundrie et al., 2003*) Spradling et al. Genetics 1999 (*Spradling et al., 1999*) Page et al. EMBO J. 2005 (*Page et al., 2005*) Luschnig et al. Genetics 2004 (*Luschnig et al., 2004*) Ashburner et al. Genetics 1990 (*Ashburner et al., 1990*) Chen et al. Curr. Biol. 2006 (*Chen et al., 2006*) Kahsai et al. G3 2018 (*Kahsai and Cook, 2018*) Lake et al. PLoS Genet. 2011 (*Lake et al., 2011*) Zhang et al. Genetics 2007 (*Zhang et al., 2007*) Maksimenko et al. Biochim Biophys Acta 2020 (*Maksimenko et al., 1863*) Zolotarev et al. BioTechniques 2019 (*Zolotarev et al., 2019*) Neumüller et al. Cell Stem Cell 2011 (*Neumüller et al., 2011*) Nazario-Yepiz et al. Mech. Dev. 2017 (*Nazario-Yepiz and Riesgo-Escovar, 2017*) Lewandowski et al. Dev. Biol. 2010 (*Lewandowski et al., 2010*) Ellis et al. PLOS ONE 2015

(*Ellis et al., 2015*). (KD: knockdown; VDRC GD; Vienna *Drosophila* Resource Center GG RNAi line; VDRC KK; Vienna *Drosophila* Resource Center KK RNAi line; TRiP: Transgenic RNAi project, Perrimon lab).

• Supplementary file 2. Multiple alignment of the ZNF domain encoded by all genes found within the *Odj-Nnk ZAD-ZNF* cluster across 20 *Drosophila* species.

• Supplementary file 3. Gene ontology analysis of transcriptional dysregulation in *Nnk-null* L1 larvae in *D. melanogaster* reveals categories of genes that are significantly over- or under-expressed upon loss of *Nnk*.

• Supplementary file 4. Nucleotide sequences of the *Nnk-mel* and *Nnk-sim* rescue transgenes. Nnk-mel rescue transgene was recoded by synonymous substitutions to make it impervious to RNAi-mediated knockdown, and the *Nnk-sim* rescue transgene was codon-optimized to match *D. melanogaster* codon preferences.

• Transparent reporting form

## Data availability

Our raw data that have been deposited to the SRA database under PRJNA644950 and PRJNA643855.

The following datasets were generated:

| Author(s) | Year | Dataset title | Dataset URL | Database and Identifier |
|---|---|---|---|---|
| Kasinathan B, Colmenares SU, McConnell H, Young JM, Karpen GH, Malik HS | 2020 | ChIP-seq of Oddjob and Nicknack | http://www.ncbi.nlm.nih.gov/bioproject/?term=PRJNA644950 | NCBI BioProject, PRJNA644950 |
| Kasinathan B, Colmenares SU, McConnell H, Young JM, Karpen GH, Malik HS | 2020 | RNA-seq of Nnk null flies | http://www.ncbi.nlm.nih.gov/bioproject/?term=PRJNA643855 | NCBI BioProject, PRJNA643855 |

The following previously published dataset was used:

| Author(s) | Year | Dataset title | Dataset URL | Database and Identifier |
|---|---|---|---|---|
| Lack JB, Cardeno CM, Crepeau MW, Taylor W, Corbett-Dettig RB, Stevens KA, Langley CH, Pool JE | 2015 | The *Drosophila* genome nexus: a population genomic resource of 623 *Drosophila melanogaster* genomes, including 197 from a single ancestral range population | https://www.ncbi.nlm.nih.gov/bioproject/PRJNA268677/ | NCBI BioProject, PRJNA268677 |

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
