## [Decision Letter]

[Editors' note: this paper was reviewed by Review Commons.]

**Acceptance summary:**

This work is significant because it challenges the widely held belief that essential genes are highly conserved over evolutionary time by demonstrating that a rapidly evolving family of transcription factors has essential functions in *Drosophila*. It also beautifully demonstrates how an essential gene in *D. melanogaster* that interacts with heterochromatin has functionally diverged from its ortholog in *D. simulans*. Overall, the study is impressive for its breadth and depth, and because it provides important new insights into how new gene functions evolve.

---

## [Author Response]

We would like to thank the three reviewers for their time and thoughtful comments on the manuscript. We found their constructive feedback extremely helpful and are grateful for the peer review process in clarifying and strengthening our study. All three reviewers were highly enthusiastic but requested a few additional experiments and clarifications. We have added the following analyses in our revision in response to their comments:

1) Two reviewers commented that polarized McDonald-Kreitman tests would provide better discrimination about whether the positive selection we saw previously in the unpolarized *D. melanogaster-D. simulans* comparison could be attributed to *D. melanogaster* alone. We have now added this analysis to our revision. Our revisiting of this analysis also made us aware of an error we had overlooked, which we have since corrected. Correcting this error has made it an even stronger correlation between genes that are evolving under positive selection and those that are required for essentiality. We corrected and revised the tables in our revision.

2) As the reviewers requested, we have also included a new Supplementary file 1, which lists the retention of all *D. melanogaster ZAD-ZNF* genes across *Drosophila* species, and the assays done to measure their phenotype.

3) Upon one reviewer’s request, we have changed the presentation of our crosses to show observed/expected ratios rather than numbers of adult flies, which are more accurate and clearer to interpret.

Reviewer #1 (Evidence, reproducibility and clarity):Kathinathan et al. is an important study with two-fold significance in general. By a thorough search of the ZAD-ZNF genes with their functional importance assayed by knockdown and knockout, an adequately big dataset in *Drosophila* was created for testing how functional essentiality evolved with the ages of genes. The reported data convincingly show that the ZAD-ZNF genes quickly evolved essential functions in viability and the positive selection is related to essentiality. Further, in a thorough experimental study, it is shown that the rapid heterochromatin-derived sequence evolution is a force to lead to dynamic evolution of the ZAD-ZNF genes with their functional importance. This provides a nice mechanistic model with strong evidence for understanding the evolution of essential functions. Nicknack and Odj are beautiful examples, with detailed analyses that reveal several unexpected insights for its role in the early development of *Drosophila*.This work, overall adding significantly novel data and new concepts to the literature on the evolution of gene functions and the evolutionary development, should be published soon in any journal with broad audiences because of the general interests of the topics and findings.There are several minor issues that can be fixed easily, with one more consideration to interpret the rescue experiment using the D. simulans orthologue.

We thank the reviewer for the positive and constructive feedback and have incorporated their comments to our revision as detailed below.

Introduction: Literature 4, a more relevant one should be cited: Lee et al., 2019. Mol Biol Evol 36: 2212-2226.

We appreciate the reviewer’s suggestion; this citation has been updated.

The recue experiment by the D. simulans orthologue gives the most intriguing result that may need more consideration of interpretation. The two hypotheses of direct and indirect effects are based on the heterochromatin of Y. There can be a third hypothesis based on the sequence difference of the coding sequences between the two species. The MK test revealed a highly excess of Dn (52, Table 2), which indeed is unusually high, given the short divergence time and size of the proteins. The male cannot be rescued while female can, also suggesting a hypothesis of rapid evolution of male functions that lead to the divergence between the two species. The further relevance of this hypothesis is that sexual selection may be a driving force, as was seen from a rate even higher than the rate driven by the adaptive evolution that helps species (both males and females) to adapt to changed environments (for example, VanKuren and Long, 2018). This is not contradictory to the two hypotheses already proposed (Pahe 17-18). After all, the Y chromosome is logically a ground where sexual selection is operating.

We are grateful to the reviewer for making this suggestion and have added this possibility and citation in our Discussion to explain the lack of male rescue.

The current analysis only counted the divergence between *D. melanogaster* and *D. simulans*, so it is unknown if the positive selection is acting on *D. melanogaster* or *D. simulans* or both. A simple counting by comparing to outgroup, *D. yakuba* and other related species, using parsimonious principle, the 52 substitutions can be assigned to the *D. melanogaster* lineage or the *D. simulans* lineage and then do MK test against the polymorphisms in the two species separately. This testing of the symmetricity of substitutions may help understand the substitution rates in the two species and better understand the rescue experiments.

We agree with this comment and have now added the polarized MK test (Table 2).

The discussion about the heterochromatin emphasizes that the heterochromatin functions are "quite variable even among close-related species". Does this mean the genomic region covered by heterochromatins also evolved quickly in the time scale of close-related species?

We now add a sentence in response to the query that “Heterochromatic DNA sequences themselves are among the most rapidly evolving component of *Drosophila* genomes” along with appropriate citations.

Materials and methods: The designed techniques in this section is tricky and cleverly designed, but the description is too concise for outsiders. For example, what was the technical purpose to make the synonymous changes? how many codons were involved? A few more words can help the audiences who are not familiar with the *Drosophila* knockdown and rescue techniques.

We thank the reviewer for this comment and suggestion. We have added additional details to the Materials and methods and Results section to more clearly explain how the experiment was performed and interpreted.

Reviewer #1 (Significance):This is a breakthrough in the area of evolution of gene functions.Reviewer #2 (Evidence, reproducibility and clarity):Authors study the evolutionary dynamics of ZAD-ZNF gene family of transcription factors in *Drosophila*. These are proteins defined by having both a ZAD (protein–protein interaction) domain and a ZNF (DNA-binding) domain. These is a family of transcription factors that has expanded in insects including flies and there are 91 proteins of this kind in *Drosophila*. However, although many are conserved and present in every fly genome, some have experienced recent duplications or losses. Some are evolving fast and others not and authors study the correlation between essentiality, mode of evolution and function. The approach is incisive because we still need to understand how often new genes become essential if at all and for what functions. I can also see how in particular some chromatin functions can be essential but evolve fast at the same time. I just would like authors to provide more details in some aspects mainly of their phylogenomics analyses and general functionality inferences.

We thank the reviewer for the excellent summary of the manuscript and kind appraisal of the work.

Major comments:1) There has been a debate about how well knockdowns recapitulate knockouts phenotype. Authors themselves mentioned some shortcomings of some knockdown approaches. Sometimes knockouts are not consistent with knockdowns. Do authors give the same weight to the knockdown and knockout data when they define essentiality? Do conclusions on essentiality hold even with smaller numbers when only knockout phenotypes are analyzed? Authors might want to provide a supplementary table containing these inferences?

We thank the reviewer for their question. We gave the knockdown and knockout data equal weight in our analysis. Indeed, our detailed analysis of the *Nnk* gene shows that the initial skepticism about the *Nnk* knockdown phenotype was unwarranted. Although it is difficult to weigh the relative strengths of available phenotypic data because of the published results, we point out that our results are completely consistent with previously published studies, and we have rigorously addressed the one discrepancy on *Nnk* essentiality. In response to the reviewer’s suggestion, we have now added a new supplementary file 1 that details the type of assay utilized to determine the essentiality of the ZAD-ZNFs.

2) Authors need to explain how they infer the age of the gene using OrthoDB and how they made sure of the losses in the genes that they classify as not universally conserved in the *Drosophila* species. Would it be possible to explain or provide a supplementary table containing the details?

We agree and apologize that we were not clear before. We have now added that gene age was determined by identifying the most divergent species that still encodes an ortholog of each *ZAD-ZNF* gene and using previous estimates of the time of divergence between these species. If orthologs were identified in basally-branching but not later-branching species by OrthoDB, we defined this as a loss of the *ZAD-ZNF* in that specific lineage.

3) Are the linker protein regions easy to align or do they have indels? Regions that do not align well could lead to the spurious detection of positive selection (Genome Res. 2011. 21(6):863-74). Have authors looked at that?

The linker protein regions between *D. melanogaster* and *D. simulans* are generally easy to align. For example, for the nucleotide sequence of *Nnk,* the linker region was 88.8% identical and had one 21 bp region that is unique to *D. simulans;* this was removed for the analysis. Linker regions get progressively more difficult to align at larger divergences, however, which is why we focused on pairwise analyses to infer positive selection. We have now appended a related manuscript file showing the high nucleotide identity between *D. simulans* and *D. melanogaster Nnk* as an example. We would be happy to append this as an additional supplementary file if the editor desires.

4) How are authors sure that duplicates are not alleles in heterozygous regions or collapsed in some species reference genome? See PLoS Comput Biol 16(7): e1008104 as an example of these effects in genome assemblies. What could be the consequence of those effects? Could this effect change any of the authors conclusions?

The reviewer is correct that our genomic inferences are sensitive to the quality of the sequencing and assembly of contigs. We have now added a sentence explaining this caveat.

5) How do authors envision that this kind of essential gene can be lost in some species?

We envision that this kind of essential gene can be lost in some species either because target(s) of the ZAD-ZNF in question were lost (e.g., regulatory sequences for heterochromatin-embedded genes) or because a paralogous ZAD-ZNF took over essential function. We have added this hypothesis in our revised Discussion.

Minor comments:1) Tables 1 and 2 do not have titles or footnotes. What do authors mean by * or the superscript numbers in the last column of Table 2?

The reviewer may have missed the title and footnotes we had originally attached along with the figure legends. We have now updated these.

Reviewer #2 (Significance):This work provides a conceptual advance. As I mentioned above, the approach is incisive because we still need to understand how often new genes become essential if at all and for what functions. It is revealing that those functions happen to be sometimes functions that evolve fast at the same time.Reviewer #3 (Evidence, reproducibility and clarity):This study analyzes the evolutionary patterns of ZAD-ZNF DNA-binding proteins in *Drosophila* and performs extensive functional characterization of two members of a particular subfamily, Nicknack and Oddjob. The authors first examine the conservation across *Drosophila* of the ZAD-ZNF genes found in D. melanogaster and use publicly available phenotypic data to ask whether the genes that show more rapid turnover across the genus or that are evolving under positive selection between *D. melanogaster* and *D. simulans* are more likely to have essential functions. Finding marginal evidence of the latter, the authors next focus on a particular cluster of tandemly duplicated ZAD-ZNF genes, which contains both Nnk and Odj. They document extensive gene gain and loss in this cluster across a wider set of species and use whole-organism RNAi to test each gene for effects on viability in *D. melanogaster*. Finding that knockdown of either Nnk or Odj results in lethality, they responsibly confirm these results using complementary forms of genetic ablation (CRISPR and transposon insertion). They then perform a series of functional experiments that show that Nnk is required for the transition to the L2 larval stage; that Odj and Nnk proteins both localize to heterochromatin, but with different specificities; and that the sequence divergence between *D. melanogaster* and *D. simulans* Nnk may specifically affect a part of the protein's function that affects male viability.Overall, this is a strong study that has well-supported conclusions. The functional analyses of Odj and Nnk clear up some ambiguities from prior papers but also break meaningful new ground, and the overall work is a nice example overall of combining evolutionary genomic and functional genetic analyses. That the authors went to the trouble of rescuing RNAi with re-coding and doing a cross-species complementation experiment is impressive and shows why this research group continues to lead the field of functional molecular evolution. The genomic documentation of ZAD-ZNF genes will be useful for other researchers studying this important gene family, as will the careful analysis of the particular cluster housing Nnk. The claims about the relative functional importance of conserved vs. fast-evolving family members are not as strong because they rely on phenotypic data in FlyBase that is, by necessity, more variable in quality, and because the genome annotations of other *Drosophila* species are of variable quality. Still, these analyses remain reasonable to include.

We thank the reviewer for the excellent summary of our manuscript and detail the responses to their constructive feedback below.

The story is acceptably complete and comprehensive as is, so I do not feel that any more experiments are necessary. However, I think there is an issue in data analysis that should be addressed. In Figures 2B, 3B and 6D, the authors report their RNAi and genetic rescue results by depicting the raw number of flies that emerge from different replicates of fertility assays. The problem in each assay is that the expected number of flies with the particular genotype of interest depends on the overall productivity of each vial's cross, which can be affected by environmental variables (e.g., food quality, presence of bacterial contamination, health of the female parent, death of any parents during the length of the assay) that vary from vial to vial. Thus, it would be more appropriate to report these data as proportions that take into account the total number of progeny that emerged from the vial and the fraction of progeny expected from Mendelian segregation patterns to be in the target class, assuming no ill effect. Crosses that show the expected number of progeny in the genetic class of interest would have value = 1, while crosses that showed complete non-viability of the genetic class of interest would have a value of 0. I don't think this will affect the major conclusions of any of the experiments in question, and thus of the study, but it would be more accurate and interpretable.

We agree with the reviewer. In all cases, we have a built-in control (siblings that inherited the Balancer chromosome instead of the Gal4 driver or knockdown transgene) that provide the expected number of progeny assuming no knockdown. As suggested by the reviewer, we now report observed/expected ratios instead of raw numbers of flies. This is the most accurate way for us to correct for any idiosyncratic differences between conditions, and the data is more interpretable.

Here are some more minor comments and suggestions:Some of the genomic data described for all members of the ZAF-ZNF family do not seem to be provided. For example, there is no supplemental table listing the species distribution of the 91 family members present in melanogaster, nor do the authors document which genes are the 28 for which experimental data show lethality or sterility. It's true that all of these data can be found on FlyBase, but it would be more convenient for other groups wishing to study other family members if the authors provided their findings here.

This suggestion was also made by reviewer #1. We now include this information in a new Supplementary file 1 in our revision.

Could the authors include a comment on the strength of the evidence for the essentiality of the 28 genes described in the Results? For example, the manuscript already notes the problem with some KK RNAi lines used by Chen et al., 2010, to study new gene viability, so if the bulk of the data that contribute to this count comes from studies like these, it would diminish the present authors' conclusions.

Reviewer #2 also raised this concern. We now provide details about the evidence for the viability phenotypes that we observed in a new Supplementary file 1. In the vast majority of cases, these data were not dependent on the KK lines (as the fly community has recognized the potential for artifact). Other knockdown sets do not have a similar systemic problem. We note especially that we do not rely on KK lines for any of the positively selected, essential *ZAD-ZNF* genes.

A couple of sentences that describe evolutionary findings could be worded more carefully. At the end of the first paragraph the authors "raise the possibility that rapid evolution of some ZAD-ZNF genes might be critical for organismal viability." I suppose the literal interpretation of this sentence is possible: if these genes hadn't changed, the ancestral organisms could have died (e.g., due to TE or repetitive region dysregulation). But, it could also be that the genes that eventually underwent positive selection already had essential functions, those functions were refined by the selection, and now modern experiments on them (which probably knocked out/down the whole gene, rather than altering the positively selected sites) have simply revealed their essential functions.

The reviewer is bringing up an important point that we have not explicitly tested whether positive selection per se is required for essentiality. However, we do show that positively selected genes are more likely to be essential. This is contrary to prevailing dogma, which argues that essential functions are more likely to be highly conserved rather than variable across species. We believe that highly unexpected finding justifies our hypothesis that rapid evolution of some ZAD-ZNF genes might be critical for organismal viability. Under the model the reviewer proposes, a previously essential gene would still have to be able to carry out its essential functions in spite of the positive selection it underwent.

Furthermore, our *D. simulans Nnk* swap experiment directly assesses the impact of positive selection. The failure of *D. simulans Nnk* to rescue male viability demonstrates the essentiality of the recent adaptive changes for proper *Nnk* function.

Likewise, the authors write that "62 of 91 ZAD-ZNFs found in *D. melanogaster* are universally retained." But didn't they find that only 73 of those genes were inferred to be present at the base of the *Drosophila* phylogeny (and thus were eligible for "universal retention")?

We would like to clarify this concern. While 73 of the genes were inferred to be present at the base of the *Drosophila* phylogeny, 11 of these genes were inferred to be lost in some species and therefore did not meet our criteria for “universal retention.” We make this clearer in our revision to avoid ambiguity.

There is a small discrepancy in the estimated time of divergence between *D. melanogaster* and *D. pseudoobscura*, listed as 35 mya but depicted as ~29 mya in Figure 1, perhaps due to different sources being used for each? The exact time is not critical for the story here, but better to be consistent (or to give a range in the text?).

We thank the reviewer for pointing out this inconsistency. We now use 30 mya as the estimated time of divergence based on the 12 *Drosophila* genomes paper.

One small weakness is that the authors did not appear to confirm that the genes of the Odj cluster in Figure 1 that did not show viability effects (CG17801 and CG17803) were being effectively knocked down, since a lack of knockdown would lead to the same result.

This is a fair point. However, we cannot address this concern due to the very low levels of expression of these genes and the absence of any tools (antibodies, transgenes) for this.

Also, it is a bit surprising that Nnk expression (and the expression of its transgenes) could not be detected by RT-PCR. I agree with the authors that the other phenotypic data supports appropriate expression of the rescue transgene, but I think they should at least discuss the possibility that differential expression levels of the D. simulans transgene in females vs. males could account for the sex-specific rescue of viability in Figure 6D.

The reviewer raises the important caveat that sex-specific rescue of viability is due to expression levels of the *D. simulans* transgene. Although this is formally possible, we think this is unlikely because the regulatory regions are shared between the *Nnk-mel* and *Nnk-sim* transgenes (and are derived from *D. melanogaster*)*; only the protein-coding regions vary.* Because the *Nnk-mel* transgene rescues both males and females, we infer that this regulatory region does not operate in a sex-specific fashion. Thus, if there was a sex-specific expression effect, it would have to be entirely mediated by the protein-coding region.

While I largely trust the Malik Lab to know heterochromatin localization patterns when they see them, Figure 6B does not have any counterstaining to show that the localization of the Nnk transgenes is actually to heterochromatin.

We appreciate the reviewer’s trust! However, to aid other readers, we highlighted the chromocenter, identified using the H3K9me staining, in figure 6B in the panels.

In Figure 3—source data 1, please clarify whether the observed/expected data shown are for one representative replicate experiment, or are combined across all 3 or 9 replicates.

We have updated the legend to clarify that the counts are combined across all replicates.

In Figure 6D, the figure itself shows male progeny as unshaded circles and female progeny as filled circles, while the legend refers to "closed circles" representing male progeny and "open circles" representing female progeny. Please check for consistency and clarity.

Thank you for politely bringing up this inconsistency. The figure and legend have been updated.

Reviewer #3 (Significance): Overall, I am impressed by this study's analytical breadth and depth of functional characterization. I think these findings will appeal to a broad audience, for example: molecular evolutionary biologists interested in the effects of adaptive evolution or gene duplication; biologists who study heterochromatin and its regulation; and, researchers who study zinc finger protein function and evolution across taxa. While previous work has catalogued ZAD-ZNF genes in insects and *Drosophila*, this study adds the important lens of adaptive evolution to this work. Likewise, while Nnk had undergone cursory, high-throughput functional testing in two prior papers (Chen et al., 2010 and Kondo et al., 2017), this study performs a much more thorough characterization and adds valuable functional comparisons to its paralog, Odj, and between the melanogaster and simulans orthologs.(For reference, my expertise lies in *Drosophila* genetic analysis, molecular evolution, and the function and evolution of novel genes. I have passing familiarity with the literature on heterochromatin and techniques like RNAseq and ChIP-seq and think those parts of this paper look reasonable, but their in-depth technical review is best left to those with more expertise.)

We appreciate the reviewer’s rigorous comments to clarify and improve the paper.

Referees cross-commenting:I appreciate point #1 raised by reviewer #2; it is a better articulated version of my second minor bullet point. Since both of us hit on this concern, I think it is a worthwhile issue for the authors to address. I also agree with their point #3, regarding potential indel variability in the linker region. I had thought of this as well, but figured that the divergence between mel and sim (the comparison most relevant for the authors' analysis of positive selection) would likely be minimal. However, I agree with reviewer #2 that it makes sense to explicitly ask the authors about this.

We appreciate this concern and have addressed it in response to reviewer #2. Briefly, linker variability is not high enough to affect the pairwise MK analyses but could affect the phylogenomic analyses, which therefore does not rely on linker alignments.

I also like reviewer #1's suggestion to attempt to polarize the MK test to determine if the adaptive evolution occurred on the branch leading to *D. melanogaster* or *D. simulans*. If data exist already for *D. yakuba* to do this analysis, I agree that it would add nicely to the study.

We agree and have added this analysis in our Revision.